# Postsynaptic Proteins at Excitatory Synapses in the Brain—Relationship with Depressive Disorders

**DOI:** 10.3390/ijms231911423

**Published:** 2022-09-28

**Authors:** Sylwia Samojedny, Ewelina Czechowska, Patrycja Pańczyszyn-Trzewik, Magdalena Sowa-Kućma

**Affiliations:** 1Students Science Club “NEURON”, Medical College of Rzeszów University, Kopisto 2a, 35-315 Rzeszow, Poland; 2Department of Human Physiology, Institute of Medical Sciences, Medical College of Rzeszów University, Kopisto 2a, 35-315 Rzeszów, Poland

**Keywords:** PSD proteins, AMPAR, NMDAR, mGluR5, NOS, Homer, depression, animal models of depression, human study, excitatory synapse

## Abstract

Depressive disorders (DDs) are an increasingly common health problem that affects all age groups. DDs pathogenesis is multifactorial. However, it was proven that stress is one of the most important environmental factors contributing to the development of these conditions. In recent years, there has been growing interest in the role of the glutamatergic system in the context of pharmacotherapy of DDs. Thus, it has become increasingly important to explore the functioning of excitatory synapses in pathogenesis and pharmacological treatment of psychiatric disorders (including DDs). This knowledge may lead to the description of new mechanisms of depression and indicate new potential targets for the pharmacotherapy of illness. An excitatory synapse is a highly complex and very dynamic structure, containing a vast number of proteins. This review aimed to discuss in detail the role of the key postsynaptic proteins (e.g., NMDAR, AMPAR, mGluR5, PSD-95, Homer, NOS etc.) in the excitatory synapse and to systematize the knowledge about changes that occur in the clinical course of depression and after antidepressant treatment. In addition, a discussion on the potential use of ligands and/or modulators of postsynaptic proteins at the excitatory synapse has been presented.

## 1. Introduction

Depressive disorders (DDs) are widespread mental illnesses worldwide and pose a significant economic and psychosocial problem. The World Health Organization (WHO) estimates that depression affects about 3.8% of the human population (with a prevalence of 5% among adults), and it increases with age (in people over 60, its frequency is 5.7%) [1,2]. It has also been observed that the lifetime risk of developing DDs is twice as high in women as in men [3]. Moreover, mental illnesses (including DDs) are important health problem among adolescents and represent one of the leading causes of disease and disability in this age group [4,5,6]. Over 50 years ago, monoamine theory, assuming that symptoms of depression were related with deficiency or imbalances of monoamines systems, i.e., serotonin, norepinephrine, dopamine, was described. To this day, drugs able to modify (increase) the brain concentration of these neurotransmitters represent the most used type of pharmacotherapy [7,8]. Unfortunately, this therapy has many disadvantages, e.g., it takes weeks or months to achieve a therapeutic effect, low remission rate, and a high risk of relapse after responding to treatment [3,9]. An additional problem is the prevalence of treatment-resistant depression (TRD), characterized as the failure to achieve an inadequate response to at least two standard antidepressants. This condition is common in clinical practice. Up to 40% of major depressive disorder (MDD) patients suffer from TRD [10,11,12,13]. These factors can discourage patients from taking antidepressants regularly or even lead to their discontinuation, significantly reducing the chance of remission and triggering several complications including suicide [14]. A better understanding of the molecular mechanisms underlying DDs, including MDD and bipolar disorder (BD), seems to be necessary to obtain new, more effective drugs (both antidepressant and anxiolytic profile) [3,15,16,17]. 

Many environmental factors contribute to the development of DDs, among which stress is particularly important [18,19,20,21]. Numerous studies on depressed patients and animal models based on stress-related procedures have shown that specific brain areas (i.e., prefrontal cortex, hippocampus, amygdala, insula) have an altered volume [22,23,24,25,26,27,28]. It’s well documented that stress factors impair the expression of neurotrophins and cause an increase in the level of the pro-inflammatory cytokines, which may result in atrophy, depleted neurogenesis, and consequently changes in neuroplasticity [21,29,30,31,32,33,34]. Moreover, it was shown that acute stress causes an increase in extracellular glutamate (Glu) levels in the hippocampus and medial prefrontal cortex (mPFC), which may lead to excitotoxicity [35,36,37]. These results also suggest that imbalance between excitatory and inhibitory neurotransmission can be a potential substrate of DDs. The truth of the above evidence is confirmed by the ever-growing interest in the role of the glutamatergic system over the decades [15,38,39,40]. A thorough understanding of the mechanisms responsible for the Glu metabolism and its influence on postsynaptic proteins at the excitatory synapse is a new research direction for achieving effective pharmacotherapy of DDs [13,15,41,42], as evidenced by the growing number of clinical studies documenting the rapid and robust antidepressant effect of ketamine (N-methyl-D-aspartate receptors (NMDAR) antagonist with additional effects on α-Amino-3-hydroxy-5-methyl-4-isoxazolepropionic acid receptors (AMPAR), hyperpolarization-activated cyclic nucleotide-gated (HCN) channels, L-type voltage-dependent calcium channel (L-VDCC), opioid receptors, and monoaminergic receptors) [43,44,45].

Glu acts through ionotropic and metabotropic receptors. However, NMDARs and AMPARs seem to be the most important for synaptic plasticity [46,47,48,49]. Synaptic plasticity, that is, changes in the onset or magnitude of long-term potentiation (LTP) or long-term depression (LTD), can be regulated by changing the number, types, or properties of these receptors in the postsynaptic membrane. AMPARs and NMDARs trafficking underlie activity-induced changes in synaptic transmission, and therefore their abundance at synapses can significantly enhance or weaken it [50,51,52,53]. The excitatory synapse is a highly dynamic structure in which receptors constantly circulate between the synaptic membrane and the cytoplasm as well as between the extra- and synaptic matrix, while postsynaptic density (PSD) proteins as well as post-translational modifications of the polypeptide chains of the receptors’ subunits play an important role in locating them and transmitting signals inside the cell [49,53,54,55,56,57]. PSD proteins modulate the signaling cascade by linking synaptic transmission from presynaptic neurons and neurotransmitter systems, mainly through by NMDARs, AMPARs, and group I metabotropic Glu receptors (especially mGluR5) [58,59,60,61]. A thorough understanding of the mechanisms responsible for the Glu turnover and its influence on postsynaptic proteins at the excitatory synapse is a new target of direction effective pharmacotherapy of DDs [41]. There is much evidence of changes in postsynaptic proteins at the excitatory synapse in depressive disorders, both in human and animal tissues. On the other hand, there are many inconsistencies in these findings. Taking this into account, the main goal of this review was to synthesize knowledge about changes in both PSD protein levels and post-translational modifications in MDD and BD patients, as well as in animal models of depression, and to determine whether these changes are characteristic of selected areas of the brain. In addition, we also reviewed the changes in these proteins following the administration of antidepressants (or compounds with antidepressant-like activity), and the expression of the genes encoding them.

## 2. The Role of Postsynaptic Density Proteins in the Excitatory Synapse

PSD is an electron-dense construction with a thickness of 20–50 nm (excitatory synapse in the hippocampus) located mainly on dendritic spines under the postsynaptic membrane, which contains many proteins that function as receptors, scaffolding and cytoskeletal elements, adhesion, signaling enzymes, and their regulators [62,63] (Figure 1). PSD is also a space whose components undergo dynamic changes, such as phosphorylation, diffusion, membrane insertion, as well as coupling and uncoupling in response to neuronal stimulation [62]. Numerous studies have shown that dysregulation within the PSD underlies many central nervous system diseases, e.g., DDs, schizophrenia, autism spectrum disorder, or neurodegenerative diseases [64,65,66].

NMDAR has a tetrameric structure and comprises two GluN1 subunits and two GluN2 subunits or, less frequently, GluN3 subunits [67] The GluN1 subunit is encoded by the *GRIN1* gene. The GluN2 subunits occur in four forms: GluN2A-D are encoded sequentially by the *GRIN2A-D* genes, and the GluN3A and GluN3B subunits are encoded by the *GRIN3A-B* genes [67,68]. The occurrence of individual subunits is dependent on various factors, e.g., brain area and developmental stage. For example, GluN2B expression already occurs in fetal life in the brain. Still, with age in humans, there is a decline in GluN2B expression in favor of GluN2A in the CA1 and CA3 regions of the hippocampus, and finally, GluN2B in adults localizes in the forebrain. In contrast, GluN2A expression is widespread in the central nervous system (CNS). GluN2C expression is only postnatal and particularly abundant in the cerebellum, and GluN2D is predominantly embryonic in the midbrain [67,69]. Each NMDAR subunit comprises an extracellular N-terminal, three transmembrane domains (M1, M3, M4), a M2 re-entrant loop that forms part of the ion channel, and an intracellular C-terminal through which the subunit interacts with other PSD proteins [69,70]. Analysis of the GluN2 C-terminal revealed that it is five-fold longer in vertebrates than in invertebrates, which may significantly affect NMDAR-mediated signaling [70]. The action on NMDAR in the limbic system is modulated, among other things, by zinc ions (Zn^2+^), whose homeostasis is dependent on zinc transporters [71]. The ubiquitous Zn^2+^ transporter 1 (ZnT-1), encoded by the *SLC30A1* gene, belongs to the solute carriers 30a (SLC30a) family and is the only transporter of this family located in the plasma membrane. ZnT-1 transports Zn^2+^ into extracellular space, thus exerting a protective effect and preventing toxicity by zinc accumulation in the cell [72]. Multiple zinc-deficiency animal models and postmortem studies have confirmed the significant role of zinc in the development of DDs [73,74].

Like NMDAR, AMPAR is also assembled as tetramer and consists of a combination of four subunits, GluA1–4 encoded by *GRIA1–4* genes [75]. The expression of individual AMPAR subunits is function-dependent and region-specific. For example, GluA2 is the brain’s most widely distributed AMPAR subunit, which translates to the occurrence in mammals of mostly AMPAR complexes consisting of: GluA1/2, GluA2/3, and GluA1/2/3 [76]. GluA2 is essential for correct AMPAR function because its deletion results in increased calcium permeability and intracellular blockade by polyamines [75,77]. The presence of the GluA2 subunit in AMPAR prevents the influx of divalent cations, such as Ca^2+^ and Zn^2+^ [78]. Within the second transmembrane domain of the GluA2 subunit, there is positively charged arginine (R) at the Q/R site, which is not encoded at the genomic level, but is generated by RNA editing. Q/R editing is performed very efficiently in most of the GluA2 subunits of mammalian neurons while the equivalent position of the other AMPAR subunits is typically preserved as glutamine (Q) in its unmodified form [79]. The presence of a positively charged R residue in GluA-containing AMPARs makes the channel impermeable to Ca^2+^, slows down the kinetics of the channel and reduces its conductivity, increases the amplitude of synaptic events, enhances neuronal excitability, and regulates AMPAR trafficking and anchoring. Since the number of calcium-permeable AMPARs in the adult brain is very small, it is expected that even a slight change in their expression will have a significant impact on synaptic transmission and the functioning of neural circuits [80]. Functional proteomic analysis showed that GluA1 and GluA2 accounted for approximately 80% of all AMPAR subunits in the hippocampus, while in the cerebellum, the GluA4 subunit was the most significant (64%). In the cortex and striatum, GluA2 is the most abundant (45%), and the proportion of GluA1 and GluA3 is estimated to be about 21–27% [81]. Each AMPAR subunit consists of four domains: extracellular N-terminal domain (NTD), ligand-binding domain (LBD), transmembrane domain that forms the ion channel (TBD), and cytoplasmic C-terminal domain (CTD), through which other PSD proteins communicate [82,83]. Interestingly, the STAR*D study found a significant association between the occurrence of suicidal thoughts in MDD patients treated with citalopram and the presence of rs4825476 (single nucleotide polymorphism) in the *GRIA3* gene [84]. 

AMPARs and NMDARs are the central receptors in the PSD that determine the proper functioning of the excitatory synapse. After an appropriately strong stimulus, Glu is released from the presynaptic neuron binds to binding sites on receptors and causes several conformational changes and ion flux, especially sodium Na^+^, potassium K^+^, and calcium Ca^2+^ [85]. NMDAR-mediated Ca^2+^ influx is responsible for a cascade of activation of many proteins, including calcium-dependent enzymes. One of these is calcium/calmodulin-dependent protein kinase II (CamKII), which acts as a signaling molecule and is crucial to modifying the actin cytoskeleton and synaptic plasticity. In mammals, CamKII is present in four isoforms: CamKIIα, CamKIIβ, CamKIIγ, CamKIIδ encoded by *CAMK2A-D* genes [86]. CamKIIα is the most common isoform in the forebrain, while CamKIIβ is particularly abundant in the cerebellum [67]. Activated CamKII contributes to the translocation of intracellular AMPARs to the PSD and phosphorylation of the GluA1 (S831) subunit, leading to enhanced synaptic transmission [75]. Additionally, CamKIIα, displaced from the cytoplasm to the PSD, also interacts with the GluN2B subunit, which is necessary for the induction of LTP [87]. 

In addition to rapid neurotransmission via ionotropic receptors, metabotropic receptors are also crucial at the excitatory synapse, among which metabotropic Glu receptor 5 (mGluR5) plays an important function in the development of DDs [88]. mGluR5 is encoded by *GRM5* gene, belongs to the family of G-protein coupled receptors (GPCRs), and, together with mGluR1, is part of group 1 and mainly localized postsynaptically [89,90]. A characteristic feature of mGluRs is their occurrence in the form of homodimers and their structure because they contain a large extracellular domain at the N-terminus called a Venus flytrap (VFT), which possesses a ligand-binding site and, via a cystine-rich domain (CRD), binds to the 7-transmembrane domain (7TM). There is a C-terminal domain in the cytoplasm of mGluR, which allows the receptor to interact with other PSD proteins [91,92]. After Glu binds to mGluR5, a series of conformational changes occur, leading to activation of the phospholipase C (PLC) pathway and production of secondary messengers, e.g., inositol 1,4,5-triphosphate (IP_3_) and diacylglycerol (DAG) consequently responsible for slow neurotransmission [93]. The effect of IP3 is a Ca^2+^ influx, which affects calcium-dependent proteins such as CamKII [90,94]. Interestingly, mGluR5 also involves the NMDAR complex, as Jin et al. showed that the application of *(RS)*3,5-dihydroxyphenylglycine (3,5-DHPG, mGluR5 agonist) resulted in increased expression of membrane GluN1 and GluN2B subunits and reduced their levels intracellularly. This effect was abolished when 3-((2-methyl-1,3-thiazol-4-yl)ethynyl)pyridine hydrochloride (MTEP, mGluR5 selective antagonist) was used, indicating an essential role for mGluR5 as a molecule that determines the movement of NMDAR subunits to the surface [94].

To maintain the correct receptors composition, a balanced scaffold with many support proteins is necessary. One of them, especially widespread in forebrain, is postsynaptic density protein 95 (PSD-95), which is responsible for stabilizing and binding other PSD proteins [67]. PSD-95 is encoded by *DLG-4* (discs large homolog 4) gene and belongs to the MAGUK (membrane-associated guanylate kinases) superfamily [65]. PSD-95 in its structure contains three PDZ (PSD-95/disc large/zonula occludens-1) domains followed sequentially by single SH3 (Src homology 3) and GK (guanylate kinase-like) domains [95]. PDZ domains allow PSD-95 to interact with many PSD proteins, including receptor (e.g., NMDAR, AMPAR, serotonin 5-HT_2_, dopamine D_2_) subunits, thus exerting a vast influence on glutamatergic, serotonergic, and dopaminergic transmission [96]. Chen et al. showed that RNA interference knockdown of PSD-95 caused PSD deficit and decreased AMPAR (but not NMDAR) levels in hippocampal neurons, thus confirming the crucial role of PSD-95 in maintaining correct protein architecture [97]. It has been observed that overexpression of PSD-95 can promote the formation of multi-innervated spines with up to seven presynaptic connections. The interaction of the PDZ2-domain of PSD-95 with nNOS (neuronal nitric oxide synthase) plays a vital role in this process [96,98]. nNOS is one of the three isoforms of the NOS enzyme and is widely distributed in the CNS and has also been demonstrated in human neutrophils [99]. Nikonenko et al. showed that both small interfering RNA (siRNA)-mediated knockdown of nNOS and pharmacological blockade by administration of L-N^G^-nitroarginine methyl ester (L-NAME, NOS inhibitor) result in the inhibition of multi-innervated spines formation, thus demonstrating the role of PSD-95 and nNOS as essential factors for synapse building [98].

The activity of individual elements included in the NMDAR-PSD-95-nNOS complex is modulated by various isoforms of the Homer 1 protein (Homer protein homolog 1) [100]. Homer 1 is widely distributed in the CNS and skeletal muscles and has nine isoforms, including Homer 1a-h and Ania-3 (Activity and neurotransmitter induced early gene 3) [101,102]. Homer-1 proteins consist of a Homer family specific EVH1 domain (enabled/vasodilator-stimulated phosphoprotein homology 1), a proline motif found in all Homer 1 proteins, and CC (coiled coil) domain, allowing the formation of homo- and heterooligomers. Short isoforms such as Homer 1a and Ania-3 lack a CC domain and function as negative modulators of Glu receptors [103]. Through the EVH1 domain, Homer 1 forms connections with other proteins, e.g., mGluRs (group 1), and can regulate their function. Wang et al. showed that Homer 1a is responsible for uncoupling between GluN2B, PSD-95, and nNOS, which was associated with the reduction of NMDAR-mediated transmission and thus had a neuroprotective effect. At the same time, Homer1 b/c facilitated mutual interactions among the NMDAR-PSD-95-nNOS complex [100]. Furthermore, an association between rs7713917 variant in the *HOMER1* gene and an increased risk of suicide attempts and a worse response to antidepressant treatment with sleep deprivation and light therapy has been shown [104,105].

Homer 1 co-occurs with Shank (SH3 and multiple ankyrin repeat domain), forming a mesh-like structure that provides a scaffold for the other PSD proteins [106]. Shank3 (SH3 and multiple ankyrin repeat domains 3) is a member of the SHANK family, abundant at the excitatory synapse, and plays a vital role in the proper maturation and formation of dendritic spines [107]. Shank3 consists of a Shank/ProSAP N-terminal (SPN) domain followed by ankyrin repeats, src homology 3 (SH3) domain, PDZ domain, and sterile alpha motif (SAM) domain. PDZ domain binds to guanylate kinase-associated protein (GKAP). It thus indirectly allows Shank3 to interact with NMDARs and AMPARs [108], while the SAM domain is located at the C-terminal and binds zinc ions [109]. Duffney et al. showed that Shank-3 knockdown using si-RNA in cortical structures resulted in decreased NMDAR-mediated ionic current and decreased GluN1 subunit expression, significantly confirming the importance of Shank3 in correct excitatory synapse function [110]. Shank3 occurs in the Shank3a-f isoforms and is expressed in the nervous system, e.g., cortex, cerebellum, amygdala, hippocampus, spinal cord, striatum, dorsal root ganglia, but also in the heart, thymocytes, and spleen [107,111]. Interestingly, in a group of seven patients with Phelan-McDermid syndrome (22q13.3 deletion, which is associated with a deletion of the *SHANK3* gene), four showed the presence of bipolar depression, which suggests a close relationship between Shank3 and this disorder [112].

## 3. Alterations in Postsynaptic Density Proteins in Depressive Disorders

Recent years, especially the last decade, have seen abundant research performed on postsynaptic proteins in the context of DDs. This review focuses only on the most important proteins, i.e., NMDAR, AMPAR, PSD-95, CamKII, Homer 1, Shank3, ZnT-1, and nNOS.

In the publications, we found clinical studies showing differences in PSD proteins expression in both depressed patients (Table 1) and preclinical studies, using animals (both mice and rats) showing depressive-like (mainly stress-induced) behaviors, as well as after treatment with drugs with antidepressant or antidepressant-like activity (Table 2, Table 3, Table 4 and Table 5). The observed alterations were sometimes varied and ambiguous, which may also be related to the diversity of the brain regions (hippocampus, prefrontal cortex, amygdala and locus coeruleus) in which the analyses were carried out. In addition, we found studies showing changes in peripheral blood cells (Table 1). Due to the large number of studies with the use of animal models, we divided our review according to the type of tested proteins (NMDAR and AMPAR, Table 2 and Table 3; other PSD proteins, Table 4 and Table 5) as well as by the species of animals used in studies (mice, Table 2 and Table 4; rats, Table 3 and Table 5). In this article, we included both research on protein and mRNA level changes.

### 3.1. Human Studies

Based on the review, it can be concluded that among the PSD proteins, NMDAR analyses in post-mortem brains of patients with DDs constitute the vast majority. Most authors indicate no changes in both the protein and mRNA levels of the GluN1 subunit [113,114,116,117,118,120].

Interestingly, these observations apply to both MDD and BD patients [116]. Although slightly contrasting results (decrease in GluN1 mRNA level) were shown by Beneyto et al. [115], different gender ratios may be of importance. Similarly, according to Gray et al., GRIN1 gene expression was higher in women with MDD, while it did not change in men. Hence, we can conclude that an important factor influencing the NMDAR proteins level is, among other things, sex [68]. Analyses of different NMDAR subunits also confirm this. As shown by Gray et al., the expression of GRIN2A and GRIN2B was higher in the group of women with MDD, while it remained unchanged in men [68]. Other authors also indicate no changes in GRIN2A expression in the MDD group with predominantly males [113,115,116]. These observations seem to be consistent, despite the different areas of the brain studied [113] or the analytical methods used [68,113,116]. On the other hand, decreased expression of GRIN2B has been observed post-mortem in the locus coeruleus of the brain, which may indicate a multifactorial etiology of DDs and the participation of the noradrenergic-glutamatergic component [113]. GRIN2A and GRIN2B expression also appears to be relatively similar in MDD and BD patients. However, when we look at the levels of these proteins in different studies, the results are more varied and inconsistent with the expression of the genes that code for them. For example, two studies [74,117] showed a decrease in GluN2A level, while subsequent studies [118,119] showed its increase. The reason for these differences seems to stem from the dissimilarity of the studied structures (prefrontal cortex vs. lateral amygdala and hippocampus), which appeared to be of secondary importance in mRNA analysis. This hypothesis may not be entirely accurate when we analyze the changes in the GluN2B protein, which show a decrease in both the prefrontal cortex and the hippocampus [117,119] and no changes in the lateral amygdala [118]. The very cause of subjects’ death in the case of protein analysis may be irrelevant, which seemed to be essential for the study of gene expression [68]. Observed discrepancies may be explained by the different functional morphology and physiology of NMDARs in other parts of the CNS. As already mentioned above, NMDARs form heterotetramers, most often composed of two GluN1 subunits that bind to two additional subunits: GluN2 (A-D), which binds Glu, or less commonly GluN3 (A-B) with high affinity for glycine [180]. NMDARs most often consist of GluN1 and GluN2 subunits, particularly GluN2A and GluN2B [181,182]. Importantly, NMDARs containing GluN2A subunits show three times faster degradation times and reduced Glu affinity compared to GluN2B-containing receptors [183]. Mellone et al. showed that ZnT-1 in hippocampal neurons binds to GluN2A (1049-1464) C-terminal (but not GluN2B) and modulates PSD-95 (postsynaptic density protein 95) activity and dendritic spike morphology [184]. Importantly, studies on MDD subjects showed an increase in the level of ZnT-1 protein, which may suggest significant changes in NMDAR signaling caused by this transporter [74]. Recent studies have shown that activation of NMDA GluN2A receptors exerts a survival-promoting effect, while NMDA GluN2B receptors, which are mainly segregated into extrasynaptic sites, show deleterious effects [185]. Two further subunits, GluN2C and GluN2D, which are known to be positively involved in synaptic transmission and working memory, are also essential [186]. In addition, it is known that NMDARs in the CNS mainly comprise triheteromeric receptors consisting of the GluN1/GluN2A/GluN2B subunits. Other triheteromeric NMDARs, including GluN1/GluN2A/GluN2D and GluN1/GluN2B/GluN2D, have been observed in the human spinal cord as well as in the rat thalamus and midbrain. Much research to date has focused on diheteromeric NMDARs containing identical GluN2 or GluN3 subunits. However, the triheteromeric NMDAR containing a combination of GluN2 and/or GluN3 subunits has different channel gating kinetics and pharmacology from diheteromeric receptors [180]. With that in mind, extending protein research on NMDARs with additional subunits and studies showing their coexistence and variety in selected regions of the brain is essential.

Like NMDARs, AMPARs are abundantly located in the postsynaptic membrane and their effects are interdependent. During LTP, presynaptic Glu release activates AMPARs, and the triggered depolarization removes the Mg^2+^ blockade of the NMDA channel and allows Ca^2+^ influx. Strong activation of NMDARs triggers the Ca^2+^-calmodulin protein kinase II (CamKII) signaling cascade, which leads to LTP, brain-derived neurotrophic factor (BDNF) secretion and synaptic amplification [185]. Human post-mortem studies of AMPAR-building proteins show much greater variability compared to NMDAR analyses. The vast majority of these studies concern gene expression [68,113,120], while one study shows changes in protein levels [74]. Two papers show no change in the GRIA1 gene expression [68,113], and one of them shows its decrease [120]. In this situation, however, attention should be paid to the structural heterogeneity of the analyzed brain tissue. On the other hand, sexual differentiation is a likely cause of heterogeneous observations in the case of GRIA2 gene expression. Duric et al. [120] in the hippocampus and Chandley et al. [113] in the locus coeruleus showed no changes in the GRIA2 level, while Gray et al. [68] showed an increase in the expression of this gene in women and no changes in the group of men. Notably, most of the groups studied in the work of Chandley et al. and Duric et al. were men [113,120]. In addition, a decrease in GRIA3 expression and no change in GRIA4 level in MDD patients has been shown [120]. The observed AMPAR subunits changes have implications for the functionality of the entire receptor. It is indicated that AMPARs being a combination of GluA1 and GluA2 are essential for the plasticity of neurons and are rapidly recycled, therefore their number in the cell membrane reflects the balance between endo- and exocytosis processes. GluA1 subunits are delivered to the synapses in an activity-dependent manner, and GluA2/3 subunits are continuously provided to synapses independently of synaptic activity. Therefore, the trafficking process is an essential mechanism underlying synaptic plasticity since the recruitment of AMPAR to the postsynaptic membrane is positively correlated with LTP, and their endocytosis negatively correlated with LTP [185].

The most crucial PSD protein, necessary to maintain the molecular organization of postsynaptic density, anchors NMDARs and AMPARs, and mediates intracellular signaling is PSD-95 [187] (Figure 1). So, it is not surprising that it has been extensively studied in the context of DDs pathophysiology. Among the studies, almost all showed a reduced level of PSD-95 protein in both suicides and patients with MDD, including those who reported suicide [74,114,117,119]. Only one study indicates an increase in PSD-95 levels, but this may be due to a different brain structure (lateral amygdala) [118]. On the other hand, studies of DLG4 gene expression indicate its decrease, and this effect may vary depending on the diagnosis (MDD vs. BD) [121].

Peripheral blood is also a source of PSD protein mRNA. It has been observed that the level of SHANK3 in peripheral blood mononuclear cells (PBMCs) correlates with treatment response among women with MDD, indicating the potential employment of SHANK3 as a marker of antidepressant response [124]. Although some studies suggest a genetic predisposition to DDs, Somani et al. showed that neutrophils of drug-naïve MDD patients significantly have higher nNOS mRNA expression, but no such correlation was found among first-degree relatives of these patients [99]. In contrast, the locus coeruleus showed reduced nNOS protein immunoreactivity, which was absent in the cerebellum [122]. Different results were observed in the lateral amygdala as there were no significant differences in nNOS protein levels in patients with MDD and adjustment disorder with depressed mood [118]. In both nNOS analyses, post-mortem toxicology studies did not show antidepressant drugs use. This is a fundamental fact, because a case-control treatment study showed that the activity of NOS in the blood was initially lower in patients with depression than in healthy controls but increased in patients who responded to treatment. In this study, no changes in NOS activity were found in depressed patients who were not receiving drugs and were not responding to treatment. There is also no correlation between the class of antidepressants used and changes in NOS activity [123]. 

### 3.2. Animal Studies

In research on the pathophysiology of depression and amid the search for new, more effective antidepressants, animal models that initiate behavior similar to depression in humans are most often used. The models based on the use of various stressors are the best validated and most frequently applicated. Among them, we find: Chronic Unpredictable Stress (CUS), Chronic Unpredictable Mild Stress (CUMS), Chronic Restraint Stress (RST), Chronic Social Defeat Stress (CSDS), and Chronic Mild Stress (CMS) [188,189]. 

The stress response in different brain regions can vary dramatically. As is already known, one of the causes of depression is neurotransmission dysfunction in the brain. Available research suggests that the glutamatergic system is involved in the pathophysiology and treatment of depression [145]. NMDARs, especially GluN2A- and GluN2B-containing, are essential for regulating neuronal plasticity [128]. Molecular analysis in the hippocampus of female mice showed an increase in the expression of Grin2A and Grin2B genes after CUS and after administration of two hormonal compounds (estradiol, progesterone) [131]. Reports suggest estrogens’ role in structural and functional synaptic plasticity and long-term potentiation (LTP) in the hippocampus [131]. In the same brain region, the levels of GluN2A, GluN2B proteins were elevated after CUMS [128]. Interestingly, the increased levels of genes encoding NMDAR proteins were demonstrated by Tamasi et al. also after venlafaxine treatment in rats [152]. On the other hand, administration of lurasidone and fluoxetine to mice and infenprodil to rats also lowered the levels of these proteins in the hippocampus, prefrontal cortex, and medial prefrontal cortex (p-GluN2B), respectively [127,145]. Moreover, the analysis of NMDA genes and proteins after 28 days of ketamine administration showed a significant decrease in the hippocampus [130]. In contrast, in the basolateral and inferior limbic prefrontal cortex, ketamine decreased the expression of only GluN2B protein [129]. Other authors also indicate that GluN2A protein levels in the hippocampus were significantly lower than GluN2B in FSL rats [146]. In turn, in a rat model of zinc deficiency, GluN2A and GluN2B protein levels were reduced, while fluoxetine treatment had no significant effect [73]. Given the above, it can be concluded that changes in NMDAR expression levels may be largely dependent on factors inducing behavioral changes and appear more critical for the antidepressant response.

AMPARs for Glu, particularly those containing the GluA1 and GluA2 subunits, also contribute to normal plasticity of neurons, and various factors can modulate their in a multidirectional manner. For example, in rats, stress increased the GRIA1 expression in the paraventricular nucleus of the hypothalamus and the level of the GluA1 in the hypothalamus [161]. On the contrary, in stressed mice, the levels of AMPAR protein (and selected subunits) were significantly lower in the medial frontal cortex [135] and the prefrontal cortex [137]. The surface receptor crosslinking of BS^3^ showed a decrease in GluA1 and GluA2 levels associated with stress in the hippocampus [139]. Emerging data suggest that some fast-acting drugs in DDs, such as scopolamine, increase Glu release and induce neurotrophic factors through AMPAR activation [137]. It was observed that the administration of fluoxetine [135] and scopolamine [137] reversed stress-induced changes. In contrast, chronic stress increased in GluA1 and p-s845-GluA1 levels in the basolateral amygdala, while fluoxetine therapy decreased their levels [136]. Chronic ketamine administration caused a decrease in AMPARs, both gene and protein levels in the whole hippocampus and CA1 region [130]. Moreover, treatment with infenprodil, fluoxetine, and S-ketamine, venlafaxine, and NaHS normalized GluA1 protein stress-induced changes in the medial prefrontal cortex and hippocampus, respectively [145,150,154,156,162]. In its turn, the application of ketamine and memantine, induced a significant increase in the pS845-GluA1 protein subunit in the hippocampus [160]. Another interesting line of research turned out to be lipopolysaccharide-induced inflammation that caused a substantial decrease in GluA1 protein in the hippocampus while the administration of ketamine and Ac-YVAD-CMK (the selective NLRP3 inflammasome inhibitor) reversed its adverse effects [134]. 

PSD proteins, such as PSD-95, CamKII, Homer 1, and Shank3 may also be involved in the development of depressive disorders. Analysis of the Dlg4 gene of stressed animals revealed a decrease in this gene in the prefrontal cortex, hippocampus, and hypothalamus in mice [167] and gene and protein increases in the hypothalamus in rats [161]. On the other hand, in the hippocampus, it showed a decrease in protein in stressed animals [166]. Interestingly, PSD-95 protein levels were significantly higher in the basolateral amygdala after stress [136]. Application of fluoxetine, asioaticoside in mice, and ketamine or sodium hydrosulfide—NaHS (CA1, CA3 region) in rats increased PSD-95 protein levels in the hippocampus [156,162,166]. In contrast, in the basolateral amygdala, fluoxetine increased protein levels [136]. Chronic administration of ketamine decreased PSD-95 protein levels in the hippocampus, while in combination with betaine it increased them [164]. In stressed rats, administration of lurasidone increased Dlg4 gene levels in the prefrontal cortex [175]. In contrast, in male mice, lurasidone and fluoxetine induced a decrease in PSD-95 after both drugs in the hippocampus and prefrontal cortex [127]. It was also demonstrated that infenprodil and YY-21 decreased protein expression after CUMS in the medial prefrontal cortex in rats [145,150]. In the RST model, imipramine administration increased PSD-95 expression in lateral nuclei and basal nuclei and decreased in medial prefrontal cortex [165]. The application of CX717 therapy increased protein levels in mPFC (2h after administration) [174]. Leem et al. also showed a decrease in p-CAMKII expression in basolateral amygdala (BA) and an increase in medial prefrontal cortex (mPFC) after drug administration [165]. CAMKII dysfunction is involved in many neurological disorders, including depression. Short-term manipulation of CAMKII can result in long-term effects on disease-related behavior. On this basis, it can cause structural changes which in turn contribute to the progression of the disease and its duration [170]. Sleep deprivation caused a decrease in the CAMKII protein subunit in the prefrontal cortex and hippocampus and administration of citalopram reversed its effects in mice [168,171]. In addition, venlafaxine treatment increased the levels of Camk2g and Camk2b genes in the frontal cortex in rats [152].

Homer 1 also has its role in synaptic plasticity. Homer genes encode a family of proteins in the PSD, where they act as multimodal adaptors. Overexpression of Homer 1a protein may contribute to decreased density of postsynaptic proteins such as Shank and inhibit postsynaptic AMPAR and NMDAR currents. Moreover, it is also involved in Glu-induced changes in the distribution of pre- and postsynaptic proteins [96]. In stressed mice, a reduction in the Homer 1a gene expression in the prefrontal cortex and protein level in both the prefrontal cortex and hippocampus were noted [128,172]. Contrary observations were made in the prefrontal cortex of stressed rats in which an increase in the Homer 1 protein was shown [177]. Furthermore, the administration of imipramine, ketamine and fluoxetine increased Homer 1a, Homer 1b/c gene expression in the cortex of mice [172].

At the cellular level, zinc is one of the main enzymes involved in biochemical processes, and disturbance of homeostasis leads to physiological or pathological problems. Despite the high demand for zinc in cells, its levels must be kept low. Among others, the ZnT protein contributes to the maintenance of zinc balance. Rafalo-Ulinska et al. showed a decrease in this protein after zinc and imipramine supplementation in the prefrontal cortex, and an increase in the hippocampus. In contrast, supplementation with zinc alone contributed to a decreased ZnT-1 protein in the prefrontal cortex [173]. 

An important signaling molecule in CNS regulating anxiety behavior is nNOS. It has been shown that nNOS-derived free radical (NO) contributes to depression caused by chronic stress [190]. nNOS activity is also coupled to NMDAR activity via the membrane-bound postsynaptic density protein PSD-95. While PSD-95 expression is inhibited, calcium ion-activated NO production via NMDAR activation is blocked, and excitotoxicity is reduced [191]. In a rat model of CUS, memantine therapy reduced nNOS protein levels [179]. Interestingly, Yin et al. demonstrated the importance of gender on nNOS protein level. In a mouse stress model, female hippocampal nNOS protein levels decreased while male increased [169]. In addition, it seems of interest to use escitalopram and Yueju-Ganmaidazao (YG), which reversed the levels of this protein in stressed animals [169]. 

## 4. Postsynaptic Density Proteins as Therapeutic Targets for DDs

Among the discussed post-synaptic proteins of the excitatory synapse, the most promising seem to be research on the possibilities of modulating glutamatergic transmission by influencing Glu receptors. Because numerous preclinical studies have shown the function of various ligands or modulators of glutamate receptors, in this paper, the authors will focus mainly on clinical trials, which give more information about the safety and effectiveness of new antidepressant therapy. Targeting pharmacotherapy of DDs to disrupt glutamatergic signaling began with using low sub-anesthetic doses of ketamine, a drug commonly used in anesthesiology [192]. The first clinical trial in patients with MDD and BD using a 40-min infusion of racemic ketamine intravenously at a subanesthetic dose of 0.5 mg/kg was conducted by Berman et al. In the ketamine group, a sustained 72 h change in scores expressed by the 25-item Hamilton Depression Rating Scale (HDRS/HAM-D) have been noticed. A the same time, in the control group, there were no significant differences [193]. Zarate et al. used the same protocol of ketamine treatment (dose, route of administration) in TRD patients. Results indicate that ketamine induced a rapid antidepressant effect (measured by the 21-item HAM-D scale and Beck Depression Inventory (BDI) that remained for seven days [194]. This finding contributed to the subsequent studies on ketamine with multiple intravenous administration in a larger group of patients. Among 97 patients with TRD, 205 ketamine intravenous infusions (0.5 mg/kg/40 min; performed 2006 and 2012) showed antidepressant activity (improvement in Montgomery–Asberg Depression Rating Scale (MADRS) scores) in 67% of TRD individuals [195]. Ketamine was also administered by intramuscular (i.m.) injection with nearly 100% bioavailability. At 0.5 mg/kg dose, there is an improved antidepressant response in MDD subjects, similar to effects observed in the group receiving electroconvulsive therapy (ECT). Moreover, oral ketamine administration at a 1 mg/kg dose (despite its relatively low bioavailability on range 20–25%) for three weeks showed pharmacological efficacy (measured by the 17-item HAM-D and Beck Scale for Suicidal Ideation (BSSI) scales) compared to ECT [196]. On the other hand, sublingual ketamine shows a slightly higher bioavailability (30%), but even more importantly, indicates an improvement in mood, sleep, and cognitive functions in 26 patients with MDD and BD. Transient light-headedness was reported side-effects of this type of treatment [197]. Moreover, the Food and Drug Administration (FDA) approved esketamine nasal spray in 2019 and it is now a promising new therapeutic option for treating TRD [198]. 

In clinical trials, other NMDAR antagonists were also used, e.g., memantine (an uncompetitive, low-affinity, and selective open-channel blocker of NMDAR). A growing body of evidence shows that memantine significantly reduced depressive symptom scores in MDD patients [199]. Moreover, limited evidence shows its effectiveness in bipolar patients as an add-on treatment [200]. Recently, the increasing number of substances targeted to regulation-specific NMDAR subunits show potential antidepressant efficacy [201]. One of them is traxoprodil (CP-101,606), an antagonist of the GluN2B subunit, the use of which in monotherapy and in combination with escitalopram, imipramine, or fluoxetine in male Swiss mice resulted in a reduction of immobility time in a forced swim test (FST) [202]. Moreover, in a group of patients using paroxetine (but not responding to selective serotonin reuptake inhibitors; SSRIs), intravenous infusion of traxoprodil (0.75 mg/kg/h for 1.5 h and then 0.15 mg/kg/h for 6.5 h) resulted in a significant reduction in MADRS scores and a higher response rate in 17-item HAM-D compared to the placebo group [203]. Furthermore, a small clinical trial involving five patients with TRD showed that oral intake 4–8 mg/day of another GluN2B antagonist, rislenemdaz (MK-0657), resulted in significant changes in 17-item HDRS and BDI scale scores compared to placebo [204]. In addition, other GluN2B subunit antagonists such as EVT-101 (also known as ENS-101) or MIJ821 are the subject of clinical trials [201,205]. 

Another strategy of experimental pharmacology of DDs is based on the use of partial NMDAR agonists. [192]. The example is D-cycloserine (DCS), a partial agonist at NMDAR-associates glycine. Heresco-Levy et al. showed that oral administration of DCS at a gradually increasing dose (up to max 1000 mg/day) resulted in significant clinical improvement as measured by the 21-item HAM-D and BDI scales in TRD patients when receiving higher doses [206]. The following partial NMDAR agonist is rapastinel (GLYX-13), which affects the glycine site of this receptor. In male Sprague-Dawley rats subjected to 14 days of adrenocorticotropic hormone (ACTH) injection, it was observed that the administration of rapastinel (10 mg/kg, i.p.) significantly reduced immobilization time in the FST but did not affect the distance traveled in the open field test (OFT) [207]. A clinical trial with intravenous rapastinel in patients with MDD showed that 5 mg/kg and 10 mg/kg doses of GLYX-13 had a significant clinical effect as measured by the 17-item HAM-D scale, which lasted from 2 h to 7 days after administration [208]. 

Potential antidepressant effectiveness was found for “ampakines”, the compounds that potentiate AMPAR function. Intraperitoneal injection of 3 mg/kg/day of S47445 decreased olfactory bulbectomy-induced motor hyperactivity in male C57BL/6J mice. Still, oral intake of 400 mg of another AMPAR positive allosteric modulator (ORG 26576) by MDD patients did not significantly produce clinical antidepressant effects [209,210]. 

Due to the involvement of mGluRs in the pathogenesis of DDs, they are also targets for new pharmacological therapy. Many of them are focused on negative allosteric modulators of mGluR5. Among them, compounds, such as basimglurant (RG-7090, RO-4917523) and AZD2066, are in clinical trials, but their effects are inconclusive and further studies are required to determine their therapeutic benefits [192,201].

## 5. Conclusions

DDs represent a serious and growing health problem among both young and older adults, and if left untreated, they can lead to a suicide death. The appearance of a series of biochemical, electrophysiological, and behavioral studies in the 1990s indicated significant changes in the glutamatergic system in depressed people which, after antidepressant treatment or ECT, contributed to the formulation of the glutamatergic theory of depression and initiated intensive research in this field. It was essential to describe the changes taking place in the Glu synapse, both in depression and after effective antidepressant therapy. The first studies focused mainly on the NMDA and AMPA ionotropic receptors, while subsequent studies considered mGluRs. Despite the large variety of available studies, taking into account the research methodology, brain region, or heterogeneity of the studied groups, it can be concluded that changes in the GluN2A and GluN2B subunits in DDs were most often identified. In the case of the AMPARs, most studies show changes in the GluA1 subunit. Therefore, it seems that these should be the most important Glu therapeutic targets for potential new antidepressant therapies. This is confirmed by clinical studies of these receptors’ new ligands/modulators (e.g., traxoprodil). Over the years, the view of excitatory synapse function has changed dramatically, such that the static structure has become highly dynamic, in which postsynaptic density proteins play an important role in both the localization and function of Glu receptors and can modify intracellular signaling. The importance of these proteins, in the development of DDs is also evidenced by numerous studies that indicate a reduced level of PSD-95, CamKII, and Homer 1 proteins. Hence, apart from the receptor mentioned above, these proteins should also become an exciting direction in the searching for more effective drugs to combat DDs.

## Figures and Tables

**Figure 1 ijms-23-11423-f001:**
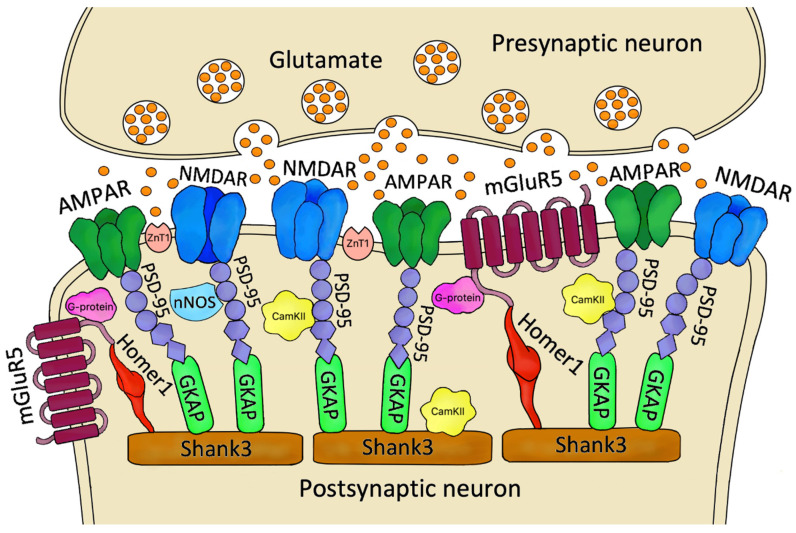
Schematic of the organization of selected PSD proteins. Released from synaptic vesicles Glu interacts with receptors on the postsynaptic neuron, specifically: AMPARs, NMDARs and mGluR5. The function of these receptors is modulated by various compounds, including nNOS, CamKII and transporters of many molecules, such as ZnT-1. An essential component of the PSD that maintains the proper proportions and alignment of the above elements are scaffold proteins, e.g., PSD-95, GKAP, Homer-1, Shank3, which bind to receptors and each other, thus performing a stabilizing function. ***Abbreviations:*** PSD—postsynaptic density; AMPAR—α-amino-3-hydroxy-5-methyl-4-isoxazolepropionic acid receptor; NMDAR—N-methyl-D-aspartate receptor; mGluR5—metabotropic Glu receptor 5; nNOS—neuronal nitric oxide synthase; CamKII—calcium/calmodulin-dependent protein kinase II; ZnT1—Zn^2+^ transporter 1; PSD-95- postsynaptic density protein 95; GKAP- guanylate kinase-associated protein; Homer-1—Homer protein homolog 1; Shank3—SH3 and multiple ankyrin repeat domains 3.

**Table 1 ijms-23-11423-t001:** Summary of clinical studies on the postsynaptic proteins in depressive disorders.

Postsynaptic Proteins	Controls [N]	Patients [N]	Samples/Brain Region	Methods	Findings	Authors’ Names
**NMDA receptor complex**						
**GluN1** **GluN2A-D** **GluN3A**	N = 19 (male = 18, female = 1)	MDD = 18 male	Locus Coeruleus (LC) Prefrontal Cortex (PFC)	qPCR	↔*GRIN1, GRIN2A* mRNA in MDD-LC ↑*GRIN2B* mRNA in MDD-LC ↔*GRIN1, GRIN2A*, *GRIN2B* mRNA in MDD-PFC	Chandley et al. [113]
**GluN1**	N = 15	MDD = 15 BD = 15	Prefrontal Cortex	Immunoautoradiography	↔GluN1 protein in MDD and BD	Toro and Deakin [114]
**GluN1** **GluN2A-D**	N = 15	MDD = 15 BD = 15	Prefrontal Cortex	In situ hybridization	↓*GRIN1* mRNA in MDD and BD ↓*GRIN2A* mRNA in MDD ↔*GRIN2B-D* mRNA in MDD ↔*GRIN2A-D* mRNA in BD	Beneyto and Meador-Woodruff [115]
**GluN1** **GluN2A-D** **GluN3A**	N = 20 (male = 11, female = 9) N = 45 (male = 20, female = 25)	MDD = 10 (male = 5, female = 4) BD = 10 (male = 5, female = 5) Suicide = 14 (male = 8, female = 6)	Prefrontal cortex	In situ hybridization	↑*GRIN2D* and ↔*GRIN1*, *GRIN2A-C*, and *GRIN3A* mRNA in MDD ↓*GRIN2C* and ↔*GRIN1*, *GRIN2A, B, D,* and *GRIN3A* mRNA in BD ↓*GRIN2B* and ↔*GRIN1*, *GRIN2A, C, D* and *GRIN3A* mRNA in Suicides	Dean et al. [116]
**GluN1** **GluN2A-B**	N = 32 (male = 19, female = 13) MDD-non-suicide (MDD-NS) = 19 (male, female)	MDD = 53 (male = 26, female = 27) MDD-Suicide (MDD-S) = 34 (male, female)	Dorsolateral Prefrontal Cortex	qPCR	↑*GRIN2B*, ↔*GRIN1* and *GRIN2A* mRNA in MDD-S-both sexes ↑*GRIN1*, *GRIN2A*, *GRIN2B* mRNA in MDD-female ↑*GRIN2B*, ↔*GRIN1* and GRIN2A mRNA in MDD-S-female ↔G*RIN1*, *GRIN2A* and GRIN2B mRNA in male MDD and MDD-S both sexes	Gray et al. [68]
**GluN2A**	N = 10 (male)	MDD = 10 (male)	Prefrontal Cortex	Western Blot	↓GluN2A protein in MDD	Rafalo-Ulinska et al. [74]
**GluN1** **GluN2A-B**	N = 14 (male = 11, female = 3)	MDD = 14 (male =11, female = 3; 8 male and 2 female were suicides)	Prefrontal Cortex	Western Blot	↔GluN1 protein in MDD ↓GluN2A, GluN2B protein in MDD	Feyissa et al. [117]
**GluN1** **GluN2A-B**	N = 14 (male = 13, female = 1)	MDD = 14 (male = 13, female = 1; 12 subjects were suicides)	Lateral amygdala	Western Blot	↔GluN1, GluN2B protein in MDD ↑GluN2A protein in MDD	Karolewicz et al. [118]
**GluN2A-B**	N = 6	Suicide victims = 17	Hippocampus	Western Blot	↑GluN2A protein in suicides ↓GluN2B protein in suicides	Sowa-Kućma et al. [119]
**GluN1** **GluN2A-D**	N = 18 (female = 7, male = 11)	MDD = 21 (female = 8, male = 13)	Hippocampus: CA1 region (CA1) and Dentate gyrus (DG)	qPCR	↔*GRIN1*, *GRIN2A-D* mRNA in CA1 and DG in MDD—both sexes	Duric et al. [120]
**AMPA receptor complex**						
**GluA1-2**	N = 19 (male = 18, female = 1)	MDD = 18 male	Locus Coeruleus (LC) Prefrontal cortex (PFC)	qPCR	↔G*RIA1*, *GRIA2* mRNA in MDD—LC and PFC	Chandley et al. [113]
**GluA1-2**	N = 32 (male = 19, female = 13) MDD-non-suicide (MDD-NS) = 19 (male, female)	MDD = 53 (male = 26, female = 27) MDD-Suicide (MDD-S) = 34 (male, female)	Dorsolateral Prefrontal Cortex	qPCR	↑*GRIA2* and ↔*GRIA1* mRNA in MDD—both sexes ↑*GRIA2* and ↔*GRIA1* mRNA in MDD-female ↔*GRIA1* and *GRIA2* mRNA in male MDD and MDD-S	Gray et al. [68]
**GluA1**	N = 10 (male)	MDD = 10 (male)	Prefrontal Cortex	Western Blot	↓GluA1 protein in MDD	Rafalo-Ulinska et al. [74]
**GluA1-4**	N = 18 (female = 7, male = 11)	MDD = 21 (female = 8, male = 13)	Hippocampus: CA1 region (CA1) and Dentate gyrus (DG)	qPCR	↓*GRIA1* and *GRIA3,* ↔*GRIA2* and *GRIA4* mRNA in MDD-both sexes-CA1 and DG	Duric et al. [120]
**Metabotropic glutamate receptor 5**						
**GluR5**	N = 32 (male = 19, female = 13) MDD-non-suicide (MDD-NS) = 19 (male, female)	MDD = 53 (male = 26, female = 27) MDD-Suicide (MDD-S) = 34 (male, female)	Dorsolateral Prefrontal Cortex	qPCR	↑*GRM5* mRNA in MDD-female ↓*GRM5* mRNA in MDD—male ↔*GRM5* mRNA in MDD-both sexes and MDD-S-both sexes	Gray et al. [68]
	N = 19 (male = 18, female = 1)	MDD = 18 male	Locus Coeruleus (LC) Prefrontal Cortex (PFC)	qPCR	↑*GRM5* mRNA in MDD—LC ↔*GRM5* mRNA in MDD—PFC	Chandley et al. [113]
**Postsynaptic density protein 95**						
**PSD-95**	N = 10 (male)	MDD = 10 (male)	Prefrontal Cortex	Western Blot	↓PSD-95 protein in MDD	Rafalo-Ulinska et al. [74]
	N = 14 (male = 11, female = 3)	MDD = 14 (male =11, female = 3; 8 male and 2 female were suicides)	Prefrontal Cortex	Western Blot	↓PSD-95 protein level in MDD	Feyissa et al. [117]
	N = 14 (male = 13, female = 1)	MDD = 14 (male = 13, female = 1; 12 subjects were suicides)	Lateral amygdala	Western Blot	↑PSD-95 protein in MDD	Karolewicz et al. [118]
	N = 6	Suicide victims = 17	Hippocampus	Western Blot	↓PSD-95 protein in suicides	Sowa-Kućma et al. [119]
	N = 15	MDD = 15 BD = 15	Prefrontal Cortex	In situ hybridization	↔*DLG4* mRNA in MDD ↓*DLG4* mRNA in BD	Kristiansen and Meador-Woodruff [121]
	N = 15	MDD = 15 BD = 15	Prefrontal Cortex	Immunoautoradiography	↓PSD-95 protein in BD vs. MDD	Toro and Deakin [114]
	N = 15	MDD = 15 BD = 15	Prefrontal Cortex	In situ hybridization	↔*DLG4* mRNA in MDD and BD	Beneyto and Meador-Woodruff [115]
	N = 20 (male = 11, female = 9) N = 45 (male = 20, female = 25)	MDD = 10 (male = 5, female = 4) BD = 10 (male = 5, female = 5) Suicide = 14 (male = 8, female = 6)	Prefrontal cortex	Western Blot	↔ PSD-95 protein in MDD and BD ↓PSD-95 protein in suicides	Dean et al. [116]
**Zinc transporters**						
**ZnT-1**	N = 10 (male) N = 8 (male, female)	MDD = 10 (male) Suicide victims = 11 (male, female)	Prefrontal Cortex	Western Blot	↑ZnT-1 protein level in MDD and suicides	Rafalo-Ulinska et al. [74]
**Nitric oxide synthases**						
**nNOS**	N = 27 (controls) N = 27 (first- degree relatives)	MDD = 29	Neutrophils (venous blood samples)	qPCR	↑*NOS1* mRNA in MDD ↔*NOS1* mRNA in first-degree relatives	Somani et al. [99]
	N = 14 (male = 13, female = 1)	MDD = 14 (male = 13, female = 1; 12 subjects were suicides)	Lateral amygdala	Western Blot	↔nNOS protein in MDD	Karolewicz et al. [118]
	N = 12 (male = 10, female = 2)	MDD = 12 (male = 9, female = 3; 12 subjects were suicides)	Locus coeruleus (LC) Cerebellum (CER)	Western blot	↓nNOS protein in MDD-LC ↔nNOS protein in MDD-CER	Karolewicz et al. [122]
**NOS**	N = 895	MDD = 460 (drug-free = 104; antidepressant group = 356: SSRI = 138; SNRI = 137; another = 81)	Blood plasma	LC-MS	↓NOS activity (L-Citrulline/L-Arginine ratio) in whole MDD group ↓NOS activity in drug-free MDD ↑NOS activity in whole MDD group at 3 months and 6 months ↑NOS activity in MDD responders group at 3 months and 6 months ↔NOS activity in MDD non-responders group at 3 months and 6 months	Loeb et al. [123]
**SH3 and multiple ankyrin repeat domains 3**						
**Shank3**	No control	MDD = 24 women BD = 32 women (All participants were treated with antidepressants and/or mood stabilizers of the 1st and 2nd generation)	Peripheral blood mononuclear cells (PBMCs)	Microarray	↑*SHANK3* mRNA in MDD after treatment	Dmitrzak-Weglarz et al. [124]

**
*Abbreviations: qPCR*
**
*—quantitative polymerase chain reaction; **LC-MS**—Liquid chromatography–mass spectrometry.*

**Table 2 ijms-23-11423-t002:** Summary of studies on the expression of postsynaptic Glu receptors (NMDAR, AMPAR and mGluR5) in mouse models of depression and/or after antidepressants treatment.

Postsynaptic Proteins	Species/Strain	Model/Treatment/Groups	Samples/Brain Region	Methods	Findings	Authors’ Names
**NMDA receptor complex**						
**NMDAR** (no information on subunits)	ICR male mice	Chronic restraint stress—3 weeks (CRS)—30 mg of Zinc (Zn)/kg of the diet After CRS—30 mg of Zn/kg of the diet + i.p. for 3 weeks: Imipramine 5 mg/kg (IMI5) Imipramine 20 mg/kg (IMI20) ZnSO_4_ 15 mg/kg (Zn15) ZnSO_4_ 30 mg/kg (Zn30) IMI (5 mg/kg) + Zn15 Imipramine was administered 1 h after Zn30 treatment.	Hippocampus	qPCR	↑NMDAR mRNA after CRS, CRS + Zn15, CRS + IMI5 and ↓NMDAR mRNA after CRS + Zn30, CRS + IMI20, and CRS + IMI5 + Zn15	Ding et al. [125]
**GluN2A**	C57BL/6J male mice	Fear Conditioning (FC)—2 weeks FC + Water (WAT) FC + Fluoxetine (FLU; 0.08 mg/mL in the drinking water) FC + WAT + Fear extinction training (EXT; 2 days) FC + FLU + EXT	Infralimbic Cortex (IL), Prelimbic cortex (PL), Basolateral Amygdala (BLA), Lateral Amygdala (LA), Central Amygdala (CEA) CA1 Strata Oriens (OR), CA1 Pyramidal (PYR), CA1 Radiatum (RAD) and CA1 Lacunosum-moleculare layers (LM)	Fluorescence Immunohistochemistry	↔ GluN2A protein in all brain region after FC + FLU ↑GluN2A protein in BLA, LA, CEA, CA1 PYR after FC+FLU+EXT	Popova et al. [126]
**GluN2A-B**	C57Bl/6J male mice	Lurasidone 3 mg/kg (LUR3) Lurasidone 10 mg/kg (LUR10) Fluoxetine 20 mg/kg (FLU20) Drugs were administered orally for 2 weeks	Hippocampus (HP) Prefrontal Cortex (PFC)	Western Blot	↓GluN2A, GluN2B protein after LUR10 and FLU20—HP and PFC	Stan et al. [127]
**GluN2A-B**	C57BL/6J male mice	Chronic Unpredictable Mild Stress (CUMS; 8 weeks) CUMS + High-frequency repetitive transcranial magnetic stimulation 15 Hz (15 Hz HF-rTMS; 4 weeks) CUMS + High-frequency repetitive transcranial magnetic stimulation 25 Hz (25 Hz HF-rTMS; 4 weeks)	Hippocampus (HP) Prefrontal Cortex (PFC)	Western Blot	↓GluN2A, GluN2B and ↔GluN1 protein level after CUMS—HP ↔GluN1, GluN2A and GluN2B protein after CUMS—PFC ↔GluN2A protein and CUMS + 25 Hz HF-rTMS and ↔GluN2B protein after CUMS+15 Hz HF-rTMS and CUMS + 25 Hz HF-rTMS—HP ↔GluN1, GluN2A, GluN2B protein after CUMS+15 Hz HF-rTMS, CUMS + 25 Hz HF-rTMS	Zuo et al. [128]
**GluN2A-B**	C57BL/6J male mice	Fear Conditioning (FC)—2 days + Ketamine 30 mg/kg i.p. (KET) acute 22 h after FC	Basolateral Amygdala (BLA) Inferior-Limbic Prefrontal Cortex (IL-PFC)	Western Blot	↔GluN2A and ↓GluN2B protein in BLA and IL-PFC after KET	Asim et al. [129]
**pS1325-GluN2A** **pS-1303-GluN2B** **GluN2A-B** **GluN1**	C57BL/6J male mice	Ketamine 30 mg i.p. 1, 3, 5, 10, 28 days (KET1, 3, 5, 10, 28, respectively) Ketamine 10, 20, 30 mg/kg for 28 days (KET_10, 20, 30, respectively)	Hippocampus	Western Blot qPCR	↔GluN1, GluN2A, GluN2B, protein after KET1, 3, 5, 10 ↓GluN2A, GluN2B and ↔GluN1 protein after KET28 ↓GluN2A and GluN2B protein after KET_10 KET_20 and KET_30 pS1325-GluN2A, pS-1303-GluN2B, GluN2A, GluN2B (total, surface, intracellular) protein after KET_30 ↓*Grin2A, Grin2B* and ↔ *Grin1* mRNA after KET_30	Luo et al. [130]
**GluN2A-B**	C57BL/6J female mice	Ovariectomy (OVX) Chronic unpredictable stress (CUS; 6 days) OVX+CUS OVX + CUS + 17β-Estradiol 0.01 mg s.c. (E) injected 19 days (every 4 days) OVX + CUS + Progesterone 0.125 mg s.c. (P) injected 19 days (every 4 days)	Hippocampus	qPCR	↑*Grin2A* and *Grin2B* mRNA after CUS *Grin2A* and *Grin2B* mRNA after OVX + CUS ↔*Grin2A* and ↑*Grin2B* mRNA after OVX + CUS + P ↑*Grin2A* and *Grin2B* mRNA after OVX + CUS+ E	Karisetty et al. [131]
**GluN2B**	Swiss male mice	Pentetrazol-induced kindling procedure (35 mg/kg; three times a week with breaks of at least 48 h—21 i.p. injections)—PTZ Valproic acid 150 mg/kg (VPA) + PTZ Pterostilbene 50 mg/kg (PTE50) + PTZ Pterostilbene 100 mg/kg(PTE100) + PTZ Pterostilbene 200 mg/kg (PTE200) + PTZ Pterostilbene and VPA were administered i.p. 30 min before each PTZ injection	Prefrontal Cortex (PFC) Hippocampus (HP)	qPCR	↔*Grin2B* mRNA level after PTE100, PTE200 + PTZ—PFC ↓*Grin2B* mRNA after VPA + PTZ—PFC ↔*Grin2B* mRNA after VPA, PTE100, 200 + PTZ in HP	Nieoczym et al. [132]
**AMPA receptor complex**						
**GluA1-pS845** **GluA1-3**	CD-1 male mice	Proteo-β-glucan from maitake 5 mg/kg (PGM5), 8 mg/kg (PGM8) and 12.5 (PGM12.5) i.p. Imipramine 15 mg/kg i.p. (IMI)	Prefrontal Cortex—whole tissue lysate Prefrontal Cortex—synaptic fraction	Western Blot	After 60 min of injection: ↔GluA1 and ↑GluA1-pS845 protein after PGM5, 8, 12.5 and IMI ↔GluA2 and GluA3 protein after PGM5, 8, 12.5 and IMI After 5 days of injection: ↑GluA1 protein after PGM 8 and 12.5 and ↑GluA1-pS845protein after PGM5, 8, 12.5 and IMI ↔GluA2 and GluA3 protein after PGM5, 8, 12.5 and IMI After 60 min of injection: ↔GluA1 and ↑GluA1-pS845 protein after PGM5, 8, 12.5 and IMI ↔GluA2 and GluA3 protein after PGM5, 8, 12.5 and IMI After 5 days of injection: ↑GluA1 and GluA1-pS845 protein after PGM5, 8, 12.5 and IMI ↑GluA2 protein after PGM8, 12.5 and IMI ↑GluA3 protein after PGM5, 8, 12.5 and IMI	Bao et al. [133]
**GluA1**	C57BL/6J male mice	Lipopolysacharide 0.8 mg/kg i.p. (LPS) Ketamine 10 mg/kg i.p. (KET) LPS + KET LPS + Ac-YVAD-CMK 8 mg/kg i.p. 30 min before LPS (YVAdD) LPS + KET (24 h after LPS) + YVAdD 8 mg/kg i.p. (30 min before LPS + KET)	Hippocampus	Western Blot	↑GluA1 protein after LPS+KET and LPS +YVAdD ↓GluA1 protein level after LPS ↔GluA1 protein level after LPS+YVAdD+KET	Li et al. [134]
**GluA1** **GluA1-pS818** **GluA1-pS831** **GluA1-pS845** **GluA2** **GluA2-pS880**	C57BL/6J male mice	Chronic restraint stress—21 days (CRS) Fluoxetine 10 mg/kg (30 min before stress)—21 days (FLU + CRS)	Medial Prefrontal Cortex	Western Blot	↔GluA1, ↓GluA1-pS831 and GluA2, and ↔GluA2-pS880 protein after CRS ↔ GluA1-pS818, ↑GluA1-pS831, ↔GluA2 and ↑ GluA2-pS880 after CRS+FLU	Park et al. [135]
**GluA1-2**	C57BL/6J male mice	Fear Conditioning (FC)—2 weeks FC + Water (WAT) FC + Fluoxetine (FLU; 0.08 mg/mL in the drinking water) FC + WAT + Fear extinction training (EXT; 2 days) FC + FLU + EXT	Infralimbic Cortex (IL), Prelimbic cortex (PL), Basolateral Amygdala (BLA), Lateral Amygdala (LA), Central Amygdala (CEA) CA1 Strata Oriens (OR), CA1 Pyramidal (PYR), CA1 Radiatum (RAD) and CA1 Lacunosum-moleculare layers (LM)	Fluorescence Immunohistochemistry	↓GluA1 protein in CA1 OR after FC+WAT+EXT ↓GluA1 protein in CA1 PYR after FC + FLU + EXT ↑GluA2 protein in all brain region after FC+WAT+EXT and FC+FLU+EXT	Popova et al. [126]
**GluA1** **GluA1-pS845** **GluA2**	C57BL/6J male mice	Chronic restraint stress—14 days (CRS) Fluoxetine 20 mg/kg i.p. 7 days after CRS (CRS + FLU)	Basolateral Amygdala	Western Blot	↑GluA1 and GluA1-pS845, and ↓GluA2 protein after CRS ↑GluA1, ↓GluA1-pS845 and ↑GluA2 protein after CRS + FLU	Yi et al. [136]
**GluA1** **GluA1-pS845**	C57BL/6J male mice	Chronic unpredictable stress 35 days (CUS) CUS + Scopolamine 10, 25 or 50 µg/kg i.p. on 34th and 35th days of CUS (CUS + SCO 10, 25, 50, respectively)		Western Blot	↓GluA1 and GluA1-pS845 protein after CUS ↑GluA1 and GluA1-pS845 after CUS + SCO 25 and 50	Yu et al. [137]
**GluA1**	Swiss male mice	Ketamine 1 or 5 mg/kg i.p. acute (KET1, 5, respectively) Guanosine (1 or 5 mg/kg i.p. acute (GUO1, 5, respectively) Corticosterone 20 mg/kg p.o.—21 days (1 week after administration KET or GUO) (CORT) Fluoxetine 10 mg/kg p.o. 3 weeks (FLU) Corticosterone 20 mg/kg p.o. 3 weeks (after FLU) (CORT) KET1 + GUO5—1 week before administration CORT—21 (CORT)	Hippocampus (HP) Prefrontal Cortex (PFC)	Western Blot	↔ GluA1 protein in Hp after KET5, GUO5 ↓GluA1 protein in HP after CORT, CORT+GUO5 ↑GluA1 protein in HP after CORT+KET ↔GluA1 protein in PFC after KET5, GUO5, CORT, CORT +KET5, CORT + GUO5 ↔GluA1 protein in HP after FLU ↓GluA1 protein in HP after CORT ↓GluA1 protein in HP after CORT+FLU ↔GluA1 protein in PFC after FLU, CORT, CORT + FLU ↔GluA1 protein in HP after KET1+GUO5 ↓GluA1 protein in HP after CORT ↓GluA1 protein in HP after CORT+KET1+GUO5 ↔GluA1 protein in PFC after KET1 +GUO5, CORT, CORT+KET1+GUO5	Camargo et al. [138]
**GluA1-2**	C57BL/6J male mice	Chronic Social Defeat Stress 10 days with CD-1 aggressor mouse (CSDS) After 10 days of CSDS, the mice were separated into susceptible and resilient subpopulations based on behavioral tests	Hippocampus	Surface receptor cross-linking with BS^3^ and Western Blot	↔GluA1, GluA2 protein in Susceptible and Resilient mice ↓GluA1, GluA2 protein after coss-linking in Susceptible mice	Li et al. [139]
**GluA1-pS845**	C57BL/6J male mice	Normal Control short hairpin RNA microinjected into PFC (NC shRNA) NC shRNA + Ketamine 10 mg/kg i.p. (KET) VGF shRNA microinjected into PFC VGF shRNA + KET	Prefrontal Cortex	Western Blot Fluorescence Immunohistochemistry	↑GluA1-pS845 protein after NC shRNA+KET ↓GluA1-pS845 protein after VGF shRNA and VGF shRNA+KET ↑GluA1-pS845 protein after NC shRNA+KET ↓GluA1-pS845 protein after VGF shRNA and VGF shRNA+KET	Shen et al. [140]
**GluA1-2**	C57BL/6J male mice	Chronic restraint stress (CRS) CRS + (2R,6R)-Hydroxynorketamine 10 mg/kg i.p. on 15 day (HNK) CRS+ HNK+ NBQX 10 mg/kg i.p. 30 min before HNK CRS + HNK + ANA-12 0.5 mg/kg i.p. at the same time as HNK	Hippocampus	Western Blot	↓GluA1,GluA2 protein level after CRS, CRS+HNK, CRS+HNK+NBQX, CRS+HNK+ANA-12	Ju et al. [141]
**pS831-GluA1** **pS880-GluA2** **GluA1-2**	C57BL/6J male mice	Ketamine 30 mg i.p. 1, 3, 5, 10, 28 days (KET1, 3, 5, 10, 28, respectively) Ketamine 10, 20, 30 mg/kg for 28 days (KET_10, 20, 30, respectively)	Hippocampus	Western BlotqPCR	↔GluA1,GluA2 protein after KET1, 3, 5, 10 and KET_30 ↓GluA1, GluA2 protein after KET_10, 20 and 30 ↓GluA1,GluA2 after KET.10, KET.20, KET.30 ↓p-S831-GluA1 and p-S880-GluA2 (total, surface, intracellular) protein after KET_30 ↓*Gria1*, *Gria2* mRNA after KET_30	Luo et al. [130]
**Metabotropic glutamate receptor 5**						
**mGluR5**	C57BL/6J male mice	Chronic Social Defeat Stress—10 days with CD-1 aggressor mouse (CSDS) After 10 days of CSDS, the mice were separated into susceptible and resilient subpopulations based on behavioral tests	Nucleus Accumbens (NAc)	Western Blot	↓mGluR5 protein in Susceptible mice ↑mGluR5 protein in Resilient mice	Xu et al. [142]
	C57BL/6J male mice	Chronic Social Defeat Stress—10 days with CD-1 aggressor mouse (CSDS) After 10 days of CSDS, the mice were separated into susceptible and resilient subpopulations based on behavioral tests Chronic restraint stress 21 days (CRS)	Hippocampus	qPCR	↑*Grm5* mRNA in Susceptible mice ↑*Grm5* mRNA after CRS	Li et al. [139]

**
*Abbreviations: qPCR*
**
*—quantitative polymerase chain reaction; **Ac-YVAD-CMK**—NLR family pyrin domain containing 3 (NLRP3) inflammasome inhibitor; **BS^3^**—bis(sulfosuccinimidyl)suberate.*

**Table 3 ijms-23-11423-t003:** Summary of studies on the expression of postsynaptic Glu receptors (NMDAR, AMPAR and GluR5) in rat models of depression and/or after antidepressants treatment.

Postsynaptic Proteins	Species/Strain	Model/Treatment/Groups	Brain Region	Methods	Findings	Authors’ Names
**NMDA receptor complex**						
**GluN2A-B**	Sprague Dawley male and female rats	Learned helpless (LH) non-learned helpless (NLH) wild type (WT)	Hippocampus	Western blot Post-embedding Immunogold	↓GluN2A/GluN2B protein in LH compared to NLH ↑GluN2B protein in LH compared to WT ↔GluN2B protein in NLH compared to WT	Bieler et al. [143]
**NMDAR** (no information on subunits)	Sprague Dawley male rats	Chronic unpredictable mild stress—3 weeks (CUMS) Paroxetine 1.8 mg/kg/d p.o. (PAR) every day from the 4th week Zn 2.3 mg/kg/d p.o. (Zn) every day from the 4th week Folic acid 21 μg/kg/d p.o. (FA) every day from the 4th week	Frontal cortex	qPCR	↓NMDAR mRNA after CUMS ↑NMDAR mRNA n after CUMS+ PAR, CUMS+ Zn+ FA, CUMS+ Zn+ FA+ PAR	Dou et al. [144]
**GluN1** **GluN2A** **pT1325-GluN2A** **GluN2B** **pS1303-GluN2B**	Sprague Dawley male rats	Chronic unpredictable stress—28 days (CUS) CUS + Ifenprodil (IFE) 3.0 mg/kg i,p. CUS + TC-DAPK 6 (an inhibitor of DAPK1) 290 nM; 0.5 μL/side, microinjected in the mPFC DAPK1 knockdown by adeno-associated virus mediated short hairpin RNA (AAV-shDAPK1), intra-PFC infusion	Medial prefrontal cortex (mPFC)	Western blot	↑GluN1, GluN2B, p-GluN2B and ↔ GluN2A, p-GluN2A protein after CUS ↔GluN2A, p-GluN2A, GluN2B and ↓p-GluN2B protein after CUS+ IFE ↑GluN1 protein after CUS+ IFE ↔GluN1 protein level after CUS+TC-DAPK 6 ↔GluN2B and ↓p-GluN2B protein level after CUS+TC-DAPK 6 ↔GluN1 protein level after CUS+ AAV-shDAPK1 ↔GluN2B protein after CUS+ AAV-shDAPK1 ↓p-GluN2B protein after CUS+ AAV-shDAPK1	Li et al. [145]
**GluN2A-B**	Flinders Sensitive Line (FSL) male rats Flinders Resistant Line (FRL) male rats	FSL FRL	Hippocampus (HP) Prefrontal cortex (PFC)	Western blot	↓GluN2A and ↑GluN2B protein in FSL—HP ↔GluN2A and GluN2B protein in FSL—PFC	Treccani et al. [146]
**GluN2A-B**	Sprague Dawley male rats	Olfactory bulbectomy (OB) Sham-operated rats (Sham) Magnesium hydroaspartate 15 mg/kg i.p. (calculated as magnesium ions; Mg15) once daily for 14 days	Prefrontal cortex (PFC) Hippocampus (HP) Amygdala (AMY)	Western blot qPCR	↔GluN2A protein after OB in PFC vs. Sham ↔GluN2A protein after OB+ Mg15 in PFC vs. OB ↔GluN2B protein after OB in PFC vs. Sham ↔GluN2B protein after OB+ Mg15 in PFC vs. OB ↔GluN2A protein after OB in HP vs. Sham ↔GluN2A protein after OB+ Mg15 in HP vs. OB ↔GluN2B protein after OB in HP vs. Sham ↔GluN2B protein after OB+ Mg15 in HP vs. OB ↔GluN2A protein after OB in AMY vs. Sham ↔GluN2A protein after OB+ Mg15 in AMY vs. OB ↔GluN2B protein after OB in AMY vs. Sham ↑GluN2B protein after OB+ Mg15 in AMY vs. OB ↔*Grin2A* mRNA after OB in PFC vs. Sham ↔*Grin2A* mRNA after OB+ Mg15 in PFC vs. OB ↔*Grin2B* mRNA after OB in PFC vs. Sham ↑*Grin2B* mRNA after OB+ 15 Mg15 in PFC vs. OB ↓*Grin2A* mRNA after OB in HP vs. Sham ↔*Grin2A* mRNA after OB+ Mg15 in HP vs. OB ↓*Grin2B* mRNA after OB in HP vs. Sham ↑*Grin2B* mRNA after OB+ Mg15 in HP vs. OB	Pochwat et al. [147]
**GluN1** **GluN2A-B**	Sprague Dawley male rats	Zinc deficiency (ZnD; 3 mg Zn/kg) for 6 weeks Zinc adequate (ZnA; 50 mg Zn/kg) Fluoxetine 10 mg/kg i,p. (FLU) after 4 weeks ZnD for subsequent 2 weeks	Hippocampus	Western blot	↑GluN1, GluN2A and GluN2B proteins after ZnD vs. ZnA ↓GluN1, GluN2A and GluN2B proteins after ZnD+ FLU vs. ZnD	Doboszewska et al. [73]
**GluN2A-B**	Sprague Dawley male rats	Chronic restraint stress—14 days (CRS)Fasudil 10 mg/kg i.p. (FAS) for 18 days from the 5th day of CRS	Hippocampus (whole tissue lysate) Hippocampus (synaptoneurosomes)	Western blot	↓GluN2A and ↔GluN2B protein after CRS ↔ GluN2A, ↔GluN2B protein after CRS+ FAS ↔GluN2A, GluN2B protein after CRS ↔GluN2A, GluN2B protein after CRS+ FAS	Román-Albasini et al. [148]
**GluN2B**	Sprague Dawley male rats	CUMS—3 weeks inta-CA1 microinjections of BDNF (0.25 μg/μL) for CUMS inta-CA1 microinjections of proBDNF (0.5 μg/μL) for naïve control	Hippocampus	Western blot	↔ GluN2B protein after CUMS ↑GluN2B protein after CUMS+BDNF ↓GluN2B protein after proBDNF	Qiao et al. [149]
**GluN2B**	Sprague Dawley male rats	CUMS—4 weeks YY-21 10 mg/kg *i.g.* once a day for 3 weeks after CUMS Fluoxetine 10 mg/kg i.p. (FLU) once a day for 3 weeks after CUMS	Medial prefrontal cortex	Western blot	↓GluN2B protein after CUMS ↑GluN2B protein after CUMS+ YY-21 ↑GluN2B protein after CUMS+ FLU	Guo et al. [150]
**GluN2A-B**	Sprague Dawley male rats	Chronic mild stress—8 weeks (CMS)	Amygdala	Western blot	↑GluN2A and GluN2B protein after CMS	Zhou et al. [151]
**GluN2A-B**	Dark Agouti male rats	Venlafaxine (VLX) 40 mg/kg s.c. (osmotic minipumps) each day for 21 days	Frontal cortex	qPCR	↑*Grin2A* and *Grin2B* mRNA after VLX	Tamási et al. [152]
**GluN2B**	Sprague Dawley male rats	Ketamine (KET) 10 mg/kg i.v. 24h before analysis	Hippocampus (HP) Nucleus accumbens (NAc) Amygdala (AMY)	Western blot	↓GluN2B protein after KET in HP ↑GluN2B protein after KET in NAc ↑GluN2B protein after KET in AMY	Piva et al. [153]
**AMPA receptor complex**						
**GluA1**	Sprague Dawley male rats	CUMS—35 days Ifenprodil 3 mg/kg i,p. (IFE) on day 37 (2 days after finish CUMS)	Hippocampus (HP) Medial prefrontal cortex (mPFC)	Western blot	↓GluA1 protein after CUMS and ↑GluA1 protein after CUMS+ IFE in HP ↓GluA1 protein after CUMS and ↑GluA1 protein after CUMS+ IFE in mPFC	Yao et al. [154]
**GluA1**	Sprague Dawley male rats	NVP-AAM07 10 mg/kg i.p.	Medial prefrontal cortex	Western blot	↑GluA1 protein after NVP-AAM077 (only 30 min after administration)	Gordillo-Salas et al. [155]
**GluA1**	Sprague Dawley male rats	CUS—28 days Ifenprodil 3.0 mg/kg i,p. (IFE) 30 min before analysis	Medial prefrontal cortex	Western blot	↓GluA1 protein after CUS ↑GluA1 protein after CUS+IFE	Li et al. [145]
**GluA1** **pS831-GluA1** **pS845-GluA1**	Sprague Dawley male rats	Olfactory bulbectomy (OB) Sham-operated rats (Sham) Magnesium hydroaspartate 15 mg/kg i.p. (calculated as magnesium ions; Mg15) once daily for 14 days	Prefrontal cortex (PFC) Hippocampus (HP) Amygdala (AMY)	Western blot	↔GluA1, pS831-GluA1, pS845-GluA1 proteins after OB in PFC vs. Sham ↔GluA1,↑pS831-GluA1 and pS845-GluA1 protein after OB+ Mg15 in PFC vs. OB ↔GluA1, pS831-GluA1 and ↑pS845-GluA1 protein after OB in HP vs. Sham ↔GluA1, pS831-GluA1 and ↓pS845-GluA1 protein after OB+ Mg15 in HP vs. OB ↔GluA1, pS831-GluA1 and pS845-GluA1 protein after OB in AMY vs. Sham ↔GluA1, pS831-GluA1 and pS845-GluA1 protein after OB+ Mg15 in AMY vs. OB	Pochwat et al. [147]
**GluA1** **pS831-GluA1** **pS845-GluA1** **GluA2**	Sprague Dawley male rats	CRS—14 days Fasudil 10 mg/kg i.p. (FAS) for 18 days from the 5th day of CRS	Hippocampus (whole tissue lysate) Hippocampus (synaptoneurosomes)	Western blot	↓GluA1 and ↔pS831-GluA1, pS845-GluA1, GluA2 protein level after CRS ↔GluA1, pS831-GluA1, pS845-GluA1, GluA2 protein after CRS+ FAS ↔GluA1, pS831-GluA1, pS845-GluA1, GluA2 protein after CRS ↔GluA1, pS845-GluA1, GluA2 and ↓pS831-GluA1 proteins after CRS+ FAS	Román-Albasini et al. [148]
**GluA3**	Dark Agouti male rats	Venlafaxine 40 mg/kg *s.c.* (osmotic minipumps) (VLX) for 21 days	Frontal cortex	qPCR	↑*Gria3* mRNA after VLX	Tamási et al. [152]
**GluA1**	Sprague Dawley male rats	CUMS—28 days S-Ketamine 20 mg/kg i,p. (S-KET) once a day for 7 days NSC23766 50 μg *i.c.v* once a day for 7 days	Hippocampus	Western blot	↓GluA1 protein after CUMS ↑GluA1 protein after CUMS+S-KET ↔GluA1 protein after CUMS+ NSC23766 + S-KET ↔GluA1 protein after CUMS + NSC23766	Zhu et al. [156]
**GluA1**	Sprague Dawley rats	Conditioned stimulus; white noise of 95 dB + unconditioned stimulus (CS-US); foot-shock 0.6 mA: 3/6/10 CS-US pairings) Ketamine 10 mg/kg i,p. (KET) 24 h before analysis Fluoxetine 10 mg/kg i,p. (FLU) for 28 days	Amygdala (AMY) Prefrontal cortex (PFC)	Western blot	↑GluA1 protein after 3 CS-US in AMY ↔GluA1 protein after 6 CS-US in AMY ↑GluA1 protein after 10 CS-US in AMY ↓GluA1 protein after 10 CS-US+ KET in AMY ↔GluA1 protein after 10 CS-US + FLU in AMY ↓GluA1 protein after 3 CS-US in PFC ↓GluA1 protein after 6 CS-US in PFC ↓GluA1 protein after 10 CS-US in PFC ↑GluA1 protein after 10 CS-US+ KET in PFC ↔GluA1 protein after 10 CS-US + FLU in PFC	Lee et al. [157]
**GluA1**	Sprague Dawley male rats	CUMS—28 days 7,8-Dihydroxy-4-methylcoumarin 10 mg/kg i,p. (Dhmc) once a day 30 min before receiving stress from day 11 to 28 for 18 days Venlafaxine 10 mg/kg i,p. (VLX) once a day 30 min before receiving stress from day 11 to 28 for 18 days	Hippocampus	Western blot	↓GluA1 protein after CUMS ↑GluA1 protein after CUMS+ VLX ↑GluA1 protein after CUMS+ Dhmc	Yang et al. [158]
**GluA1** **pS845-GluA1**	Sprague Dawley male rats	CUMS—28 days YY-21 10 mg/kg *i.g.* once a day for 3 weeks after CUMS Fluoxetine 10 mg/kg i,p. (FLU) once a day for 3 weeks after CUMS	Medial prefrontal cortex	Western blot	↔GluA1 and ↓pS845-GluA1 proteins after CUMS ↔GluA1 and ↑pS845-GluA1 proteins after CUMS+ YY-21 ↔GluA1 and ↑pS845-GluA1 proteins after CUMS+ FLU	Guo et al. [150]
**GluA1**	Sprague Dawley male rats	Zinc hydroaspartate 5 mg/kg i,p. (Zn) 30 min, 3 h and 24 h before decapitation	Prefrontal cortex	Western blot	↔GluA1 protein 30 min after Zn5 treatment ↑GluA1 protein 3 h after Zn5 treatment ↑GluA1 protein 24h after Zn treatment	Szewczyk et al. [159]
**GluA1** **pS845-GluA1**	Sprague Dawley male and female rats	Ketamine 10 mg/kg i,p. (KET) before analysis Memantine 10 mg/kg i,p. (MEM) before analysis	Hippocampus	Western blot	↑GluA1 and pS845-GluA1 protein after KET ↔GluA1 and ↑pS845-GluA1 protein after MEM	Zhang et al. [160]
**GluA1**	Sprague Dawley male rats	CUS—21 days	Hypothalamic paraventricular nucleus (hPVN) Hypothalamus (HY)	Western blot qPCR	↑ GluA1 protein after CUS in hPVN ↑GluA1 protein and ↑*Gria1* mRNA after CUS in HY	Li et al. [161]
**GluA1-2**	Sprague Dawley male rats	CUMS—35 days sodium hydrosulfide 11.2 mg/kg i.p. (NaHS) before behavioral test	Hippocampus	Western blot	↓GluA1 and GluA2 protein after CUMS ↑GluA1 protein after CUMS+ NaHS ↑GluA2 protein after CUMS+ NaHS	Hou et al. [162]
**GluA1**	Sprague Dawley male rats	Ketamine 10 mg/kg i.v. (KET) 24h before sacrifice	Hippocampus (HP) Nucleus accumbens (NAc) Amygdala (AMY)	Western blot	↓GluA1 protein after KET in HP ↓GluA1 protein after KET in NAc ↔GluA1 protein after KET in AMY	Piva et al. [153]
**Metabotropic glutamate receptor 5**						
**mGluR5**	Sprague Dawley male rats	CUMS—42 days Saikosaponin D: high-dose 1.5 mg/kg/d p.o. (SSDH); low-dose 0.75 mg/kg/d p.o. (SSDL) for 3 weeks Fluoxetine 2.0 mg/kg/d p.o. (FLU) for 3 weeks	Hippocampus	Western blot Immunohistochemistry qPCR	↓mGluR5 protein after CUMS ↑mGluR5 protein after CUMS+ FLU ↑mGluR5 protein after CUMS+ SSDH ↑mGluR5 protein after CUMS+ SSDL ↓mGluR5 protein after CUMS ↑mGluR5 protein after CUMS+ FLU ↑mGluR5 protein after CUMS+ SSDH ↑mGluR5 protein after CUMS+ SSDL ↓*Grm5* mRNA after CUMS ↑*Grm5*mRNA after CUMS+ FLU ↑*Grm5* mRNA after CUMS+ SSDH ↑*Grm5* mRNA after CUMS+ SDDL	Liu et al. [163]
**mGluR5**	Sprague Dawley male rats	Ketamine 10 mg/kg i.v. (KET) 24h before sacrifice	Hippocampus (HP) Nucleus accumbens (NAc) Amygdala (AMY)	Western blot	↓mGluR5 protein after KET in HP ↓mGluR5 protein after KET in NAc ↑mGluR5 protein after KET in AMY	Piva et al. [153]

***Abbreviations: YY-21—****novel furostan skeleton secondary timosaponin, was obtained from timosaponin B-III via hydrolysis with hydrochloric acid; **NVP-AAM07**—[(S)-[[(1S)-1-(4-bromophenyl)ethyl]amino]-(2,3-dioxo-1,4-dihydroquinoxalin-5-yl)methyl]phosphonic acid; preferring GluN2A subunit antagonist; **NSC23766—**(*N6*-[*2*-(*4*-Diethylamino-1-methyl-butylamino)-*6*-methyl-pyrimidin-4-yl]-2-methyl-quinoline-4,*6*-diamine); Rac1 inhibitor; **qPCR**—quantitative polymerase chain reaction.*

**Table 4 ijms-23-11423-t004:** Summary of studies on the expression of postsynaptic density proteins in mouse models of depression and/or after antidepressants treatment.

Postsynaptic Proteins	Species/Strain	Model/Treatment/Groups	Brain Region	Methods	Findings	Authors’ Names
**Postsynaptic density protein 95**						
**PSD-95**	ICR male, female mice	Ketamine 20 mg/kg i,p. 14 days (twice a day) (KET) Betaine 0, 30, 100 mg/kg i,p. 7 days after KET (BET30, 100)	Hippocampus—CA1	Immunofluorescence	↓PSD-95 protein after KET ↑PSD-95 protein after BET100, KE+BET30 and KE+BET100	Chen et al. [164]
	Swiss male mice	Ketamine 1 or 5 mg/kg i,p. acute (KET1, 5, respectively) Guanosine (1 or 5 mg/kg i,p. acute (GUO1, 5, respectively) Corticosterone 20 mg/kg p.o.—21 days (1 week after administration KET or GUO) (CORT) Fluoxetine 10 mg/kg p.o. 3 weeks (FLU) Corticosterone 20 mg/kg p.o. 3 weeks (after FLU) (CORT) KET1 + GUO5—1 week before administration CORT—21 days (CORT)	Hippocampus (HP) Prefrontal Cortex (PFC)	Western Blot	↔PSD-95 protein in HP after KET5, GUO5↓PSD-95 protein in HP after CORT, CORT+GUO5 ↑PSD-95 protein in HP after CORT+KET5 ↔PSD-95 protein in PFC after KET5, GUO5, CORT, CORT+KET5, CORT+GUO5 ↔PSD-95 protein in HP after FLU ↓PSD-95 protein in HP after CORT ↓PSD-95 protein in HP after CORT +FLU ↔PSD-95 protein in PFC after FLU, CORT, CORT + FLU ↔PSD-95 protein in HP after KET1+GUO5 ↓PSD-95 protein in HP after CORT, CORT+KET1+GUO5 ↔PSD-95 protein in PFC after KET1 + GUO5, CORT, CORT + KET1 + GUO5	Camargo et al. [138]
	C57BL/6 male mice	Chronic restraint stress (CRS)—21 days CRS + Imipramine 20 mg/kg i.p. 21 days 30 min before stress (IMI)	Basal nuclei (BLA) Lateral nuclei (LAT) Medial Prefrontal Cortex (mPFC)	Immunohistochemistry	↑PSD-95 protein in LAT and BLA after CRS ↓PSD-95 protein in LAT and BLA after CRS+IMI ↑PSD-95 expression in mPFC after CRS+IMI ↓PSD-95 expression in mPFC after CRS	Leem et al. [165]
	C57BL/6 male mice	Chronic Unpredictable Mild Stress (CUMS)—8 weeks CUMS + Fluoxetine 20 mg/kg were administered by oral gavage 4 weeks (4th-8th week of the experiment) (FLU) CUMS + Asiaticoside 10, 20 or 40 mg/kg were administered by oral gavage 4 weeks (4th-8th week of the experiment) (ASI10, 20, 40, respectively)	Frontal Cortex Hippocampus	Western Blot	↔PSD-95 protein after CUMS, CUMS + FLU, CUMS+ASI10, 20, 40 ↓PSD-95 protein after CUMS ↑PSD-95 protein after CUMS+ASI10, CUMS + ASI20, CUMS +ASI40, CUMS+FLU	Luo et al. [166]
	C57Bl/6J male mice	Lurasidone 3 mg/kg (LUR3) Lurasidone 10 mg/kg (LUR10) Fluoxetine 20 mg/kg (FLU20) Drugs were administered orally for 2 weeks	Hippocampus (HP) Prefrontal Cortex (PFC)	Western Blot	↓PSD-95 protein after LUR10 and FLU20 ↓PSD-95 protein after LUR10 and FLU20	Stan et al. [127]
	C57Bl/6J male mice	CRS—14 days Fluoxetine 20 mg/kg i,p. 7 days after CRS (FLU)	Basolateral Amygdala	Western Blot	↑PSD-95 protein after CRS ↓PSD-95 protein after CRS + FLU	Yi et al. [136]
	C57BL/6J male mice	CRS CRS + (2R,6R)-Hydroxynorketamine 10 mg/kg i,p. on 15 day (HNK) CRS+ HNK+ NBQX 10 mg/kg i,p. 30 min before HNK CRS + HNK + ANA-12 0.5 mg/kg i,p. at the same time as HNK	Hippocampus	Western Blot	↓PSD-95 protein after CRS, CRS+HNK, CRS+HNK+NBQX, CRS+HNK+ANA-12	Ju et al. [141]
	KM (Kunming) male mice	UCMS—8 weeks UCMS +Alarin 1 nmol *i.c.v.* (Ala1) UCMS + Alarin 2 nmol *i.c.v.* (Ala2) UCMS + Ala2 + Rapamycin 3 mg/kg i,p. (Rapa)	Prefrontal Cortex (PFC) Hippocampus (HP) Hypothalamus (HY) Olfactory Bulb (AL)	qPCR Western Blot	↓*Dlg4* mRNA after UCMS, UCMS + Ala2+ Rapa in PFC, HP, Hy, AL. ↑*Dlg4*mRNA after UCMS+ Ala1 Ala2 in PFC, HP ↔*Dlg4* mRNA after UCMS + Ala1 in HY, AL ↑*Dlg4* mRNA after UCMS + Ala2 in HY, AL ↓PSD-95 protein after UCMS, UCMS+Ala2+Rapa in PFC ↑PSD-95 protein after UCMS+Ala2 in PFC ↔PSD-95 protein after UCMS+Ala1 in PFC ↓PSD-95 protein after UCMS, UCMS + Ala2 + Rapa in Hp ↑ PSD-95 protein after UCMS+Ala1, Ala2 in Hp ↓PSD-95 protein after UCMS in Hy ↔PSD-95 protein after UCMS + Ala1, Ala2, Ala2 + Rapa in HY ↓PSD-95 protein after UCMS in AL ↑PSD-95 protein after UCMS + Ala2 in AL ↔PSD-95 protein after UCMS + Ala1, Ala2+Rapa	Zhuang et al. [167]
**Calcium/calmodulin-dependent protein kinase II**						
**p-CaMKII**	C57BL/6 male mice	CRS—21 days CRS + Imipramine 20 mg/kg i,p. 21 days 30 min before stress (IMI)	Basal nuclei (BA) Lateral nuclei (LAT) Medial Prefrontal Cortex (mPFC)	Immunohistochemistry	↑p-CamKII protein in LAT, BA after CRS and ↓after CRS+IMI ↓p-CamKII protein after CRS in mPFC ↑p-CamKII protein after CRS+IMI in mPFC	Leem et al. [165]
**p-T286-CaMKII** **t-CaMKII**	C57BL/6J male, female mice	Citalopram 10 mg/kg i,p. 5 days (CTM) Sleep Deprivation 24h (SD) CTM+SD	Prefrontal Cortex	Western Blot	↓CaMKII-p-T286 and ↔ t-CaMKII protein after SD ↑CaMKII-p-T286 protein after CTM and CTM + SD	Misrani et al. [168]
**CaMKII**	ICR male, female mice	Yueju-Ganmaidazao^ 1 g/kg *i.g.* 4 days (YG) Escitalopram 10 mg/kg *i.g.* 4 days (ES) Lipopolysaccharide 1 mg/kg i,p. 4 days (LPS) LPS + YG 4 days LPS + ES 4 days Chronic Unpredictable Stress 3 weeks (CUS)+ YG CUS + ES	Hippocampus	Western Blot	↔CaMKII protein after YG, ES, YG+LPS, ES+LPS (both sexes) ↓CaMKII (female) and ↑CaMKII (male) protein after CUS ↑CaMKII (female) and ↓CaMKII (male) protein after CUS+YG, CUS+ES	Yin et al. [169]
**p-CaMKII** (no information about the site of phosphorylation)	ICR male, female mice	Chronic Learned Helplessness 13 days (training: 1–3, 8, 13 day) (cLH) Yueju 1.25 g/kg in 0.9% saline for 21 days after cLH (YJ) Fluoxetine 18 mg/kg in 0.9% saline for 21 days after cLH (FLU)	Hippocampus	Western Blot	↓p-CAMKII protein after cLH and cLH+YJ (at day 7) ↔ p-CAMKII protein after cLH+FLU (at day 7) ↓p-CAMKII protein after cLH and cLH+FLU (at day 26—after 5 days break in drug administration)	Zou et al. [170]
**p-T286- CaMKII**	C57BL/6J male, female mice	Citalopram 10 mg/kg i,p. 5 days (CTM) Sleep Deprivation 24h (SD) CTM+SD	Hippocampus	Western Blot	↓p-T286-CAMKII protein after SD ↑p-T286-CAMKII protein after CTM, CTM + SD	Misrani et al. [171]
**CaMKIIα**	C57BL/6J male mice	Fear Conditioning (FC)—2 days + Ketamine 30 mg/kg i,p. (KET) acute 22 h after FC	Basolateral Amygdala (BLA) Inferior-Limbic Prefrontal Cortex (IL-PFC)	Western Blot	↔CAMKIIα protein after KET in BLA and IL-PFC	Asim et al. [129]
**Homer scaffold protein 1**						
**Homer 1a**	C57BL/6J male mice	Chronic Unpredictable Mild Stress (CUMS; 8 weeks) CUMS + High-frequency repetitive transcranial magnetic stimulation 15 Hz (15 Hz HF-rTMS; 4 weeks) CUMS + High-frequency repetitive transcranial magnetic stimulation 25 Hz (25 Hz HF-rTMS; 4 weeks)	Hippocampus (HP) Prefrontal Cortex (PFC)	Western Blot	↓Homer 1a protein in HP and PFC after CUMS ↔Homer 1a protein in HP and PFC after CUMS+15 Hz HF-rTMS, CUMS + 25 Hz HF-rTMS	Zuo et al. [128]
**Homer 1a** **Homer 1b/c**	C57BL/6 female mice	Chronic behavioral despair model—5 days (CDM5) CDM (for 4 weeks after CDM5 without any antidepressant treatment) Imipramine 15 mg/kg in water for 4 weeks after CDM5 (IMI) Fluoxetine 15 mg/kg for 4 weeks in water after CDM5 (FLU) Ketamine 3 mg/kg i.p. (KET) after CDM5 Positive control; single injection in 32th day experiments	Cortex	qPCR	↓*Homer1a* and *Homer1b/c* mRNA expression after CDM ↑*Homer1a* and *Homer1b/c* mRNA expression after IMI, FLU, KET	Sun et al. [172]
**Homer 1a** **Homer 1b/c**	C57BL/6 J, CD1 male mice	Chronic Social Defeat Stress 10 days with CD-1 aggressor mouse (CSDS) After 10 days of CSDS, the mice were separated into susceptible and resilient subpopulations based on behavioral tests	Medial Prefrontal Cortex (mPFC) Hippocampus (HP) Amygdala (AMY)	qPCR Western Blot	↔*Homer1a* mRNA in Susceptible, and Resilient mice after CSDS—all brain regions ↑*Homer1b/c* mRNA in Susceptible mice after CSDS—HP ↔Homer 1a protein in Susceptible and Resilient mice after CSDS—all brain regions ↑Homer 1b/c protein in Susceptible mice after CSDS—HP	Li et al. [139]
**Zinc transporters**						
**ZnT-1**	Albino Swiss male mouse	Zn 40 mg/kg p.o. (Zn) Imipramine 60 mg/kg p.o.(IMI) Zn + IMI Mice were decapitated sixty minutes following drug administration.	Prefrontal Cortex Hippocampus	Western Blot	↓ZnT-1 protein after Zn ↔ZnT-1 protein after IMI and Zn + IMI ↓ZnT-1 protein after Zn ↔ ZnT-1 protein level after IMI ↑ ZnT-1 protein level after Zn + IMI	Rafało-Ulińska et al. [173]
**Nitric oxide synthases**						
**nNOS**	ICR male, female mouse	Yueju-Ganmaidazao 1 g/kg by gavage 4 days (YG) Escitalopram 10 mg/kg by gavage 4 days (ES) Lipopolysaccharide 1 mg/kg i.p. 4 days (LPS) LPS + YG 4 days LPS + ES 4 days Chronic Unpredictable Stress 3 weeks (CUS)+ YG CUS + ES	Hippocampus	Western Blot	↔ nNOS protein after YG, ES, YG+LPS, ES+LPS (both sexes) ↓nNOS protein after CUS (female) ↑nNOS protein after CUS+YG, CUS+ES (female) ↑nNOS protein after CUS (male) ↓nNOS protein after CUS+YG, CUS+ES (male)	Yin et al. [169]

**
*Abbreviations: Asiaticoside*
**
*—product derived from the plant Centella asiatica. It has been shown to possess wound healing, anti-inflammatory and liver protective effects. In addition, studies demonstrated that asiaticoside could attenuate neurobehavioral andneurochemical; **Alarin**—neuropeptide with antidepressant activity; **Yueju-Ganmaidazao (YG)**–an herbal medicine prescribed for the treatment of mood disorders, consisting of two classic traditional Chinese herbal medicines Yueju and Ganmaidazao. Yueju and Ganmaidazao are used to treat depression.*

**Table 5 ijms-23-11423-t005:** Summary of studies on the expression of postsynaptic density proteins in rat models of depression and/or after antidepressants.

Postsynaptic Proteins	Species/Strain	Model/Treatment	Brain Region	Methods	Findings	Authors’ Names
**Postsynaptic density protein 95**						
**PSD-95**	Sprague Dawley male rats	CUS—28 days Ifenprodil 3.0 mg/kg i,p. (IFE) 30 min before analysis	Medial prefrontal cortex	Western blot	↓PSD-95 protein after CUS ↑PSD-95 protein after CUS+ IFE	Li et al. [145]
	Sprague Dawley male rats	CUMS—35 days sodium hydrosulfide 11.2 mg/kg i,p. (NaHS) before behavioral test	Hippocampus (CA1 and CA3 region) Dentate gyrus	Western blot	↓PSD-95 protein after CUMS in all regions ↑PSD-95 protein after CUMS+ NaHS in all regions	Hou et al. [162]
	Sprague Dawley male rats	CUS—21 days	Hypothalamus	Western blot qPCR	↑PSD-95 protein after CUS ↑*Dlg4* mRNA after CUS	Li et al. [161]
	Sprague Dawley male rats	CX717 20 mg/kg i,p. 24 h after the pretest	Medial prefrontal cortex	Western blot	↑PSD-95 protein after CX717 (only 2 h after administration)	Gordillo-Salas et al. [174]
	Sprague Dawley male rats	CUMS—28 days S-Ketamine 20 mg/kg i,p. (S-KET) once a day for 7 days NSC23766 50 μg *i.c.v* once a day for 7 days	Hippocampus	Western blot	↓PSD-95 protein after CUMS ↑PSD-95 protein after CUMS+ S-KET ↑PSD-95 protein after CUMS+ NSC23766 + S-KET ↑PSD-95 protein after CUMS+ NSC23766	Zhu et al. [156]
	Sprague Dawley male rats	Conditioned stimulus; white noise of 95 dB + unconditioned stimulus (CS-US); foot-shock 0,6 mA: 3/6/10 CS-US pairings) Ketamine 10 mg/kg i,p. (KET) 24h before analysis Fluoxetine 10 mg/kg i,p. (FLU) for 28 days	Amygdala (AMY) Prefrontal cortex (PFC)	Western blot	↑PSD-95 protein after 3 CS-US in AMY ↔PSD-95 protein after 6 CS-US in AMY ↑PSD-95 protein after 10 CS-US in AMY ↔PSD-95 protein after 10 CS-US+ KET in AMY ↔PSD-95 protein after 10 CS-US+ FLU in AMY ↓PSD-95 protein after 3 CS-US in PFC ↓PSD-95 protein after 6 CS-US in PFC ↓PSD-95 protein after 10 CS-US in PFC ↑PSD-95 protein after 10 CS-US+ KET in PFC ↔PSD-95 protein after 10 CS-US + FLU in PFC	Lee et al. [157]
	Sprague Dawley male rats	CUMS—28 days YY-21 10 mg/kg *i.g.* once a day for 3 weeks after CUMS Fluoxetine 10 mg/kg i,p. (FLU) once a day for 3 weeks after CUMS	Medial prefrontal cortex	Western blot	↓PSD-95 protein after CUMS ↑PSD-95 protein after CUMS+ YY-21 ↔PSD-95 protein after CUMS+ FLU	Guo et al. [150]
	Sprague Dawley male rats	Zinc hydroaspartate 5 mg/kg i,p. (Zn) 30 min, 3 h and 24 h before decapitation	Prefrontal cortex	Western blot	↔PSD95 protein 30min after Zn treatment ↑PSD95 protein level 3 h after Zn treatment ↔PSD95 protein level 24h after Zn treatment	Szewczyk et al. [159]
	Wistar male rats	Chronic mild stress—7 weeks (CMS) Lurasidone (LUR) 1mL/kg p.o. for 5 weeks	Prefrontal cortex	qPCR	↓*Dlg4* mRNA after CMS (stress-reactive group) ↑*Dlg4*mRNA after CMS+ LUR	Luoni et al. [175]
**PSD-95**	Sprague Dawley male rats	CUMS—42 days Saikosaponin D: high-dose 1.5 mg/kg/d p.o. (SSDH); low-dose 0.75 mg/kg/d p.o. (SSDL) for 3 weeks Fluoxetine 2.0 mg/kg/d p.o. (FLU) for 3 weeks	Hippocampus	Western blot Immunohistochemistry qPCR	↓PSD-95 protein after CUMS ↑PSD95 protein CUMS+ FLU ↑PSD95 protein CUMS+ SDDH ↔PSD95 protein CUMS+ SDDL ↓PSD-95 protein after CUMS ↑PSD95 protein CUMS+ FLU ↑PSD95 protein CUMS+ SDDH ↔PSD95 protein CUMS+ SDDL ↓*Dlg4* mRNA after CUMS ↑*Dlg4* mRNA after CUMS+ FLU ↑*Dlg4* mRNA after CUMS+ SDDH ↑*Dlg4* mRNA after CUMS+ SDDL	Liu et al. [163]
**Calcium/calmodulin-dependent protein kinase II**						
**CaMKII** **p-T286-CaMKII** **p-T286-CaMKII** **α** **p-T286-CaMKII** **β**	Sprague Dawley male rats	CUMS—30 days GXDSF (*Salviae miltiorrhizae Radix et Rhizoma*, the *Notoginseng Radix et Rhizoma* and the oil extract of Dalbergiae odoriferae Lignum) 1-fold (L); 2-fold (M); 4-fold (H) *i.g.* once a day for 43 days	Hippocampus	Western blot	↔CaMKII protein after CUMS ↔CaMKII protein after CUMS+ GXDSF in all groups ↑p-CaMKII protein after CUMS ↓p-CaMKII protein after CUMS+ GXDSF in all groups ↑p-CaMKIIα protein after CUMS ↓p-CaMKIIα protein after CUMS+ GXDSF in all groups ↑p-CaMKIIβ protein after CUMS ↓ p-CaMKIIβ protein after CUMS+ GXDSF in all groups	Xie et al. [176]
**CaMKIIγ** **CaMKII** **β**	Dark Agouti male rats	Venlafaxine (VLX) 40 mg/kg *s.c.* (osmotic minipumps) each day for 21 days	Frontal cortex	qPCR	↑*Camk2g* mRNA after VLX ↑*Camk2b* mRNA after VLX	Tamási et al. [152]
**Homer scaffold protein 1**						
**Homer 1**	Wistar male rats	CMS (8 weeks): (1) Anhedonic-like: >30% within-subject decrease in sucrose intake, (2) Stress-resilient: <10% within-subject decrease in sucrose intake.	Prefrontal cortex	Western blot	↑Homer1 protein in resilient group compared to anhedonic	Palmfeldt et al. [177]
**Homer 1**	Sprague Dawley male rats	CRS—14 days Fasudil 10 mg/kg i,p. (FAS) for 18 days from the 5th day of CRS	Hippocampus (synaptoneurosomes)	Western blot	↔Homer1 protein after CRS ↔Homer1 protein after CRS+ FAS	Román-Albasini et al. [148]
**Homer 1**	Sprague Dawley male rats	CUMS—42 days Saikosaponin D: high-dose 1.5 mg/kg/d p.o. (SSDH); low-dose 0.75 mg/kg/d p.o. (SSDL) for 3 weeks Fluoxetine 2.0 mg/kg/d p.o. (FLU) for 3 weeks	Hippocampus	Western blot Immunohistochemistry qPCR	↑Homer1b/c protein after CUMS ↓Homer-1b/c protein after CUMS+ FLU ↓Homer1b/c protein after CUMS+ SSDH ↓Homer1b/c protein after CUMS+ SSDL ↑Homer1b/c protein after CUMS ↓Homer1b/c protein after CUMS+ FLU ↓Homer1b/c protein after CUMS+ SSDH ↓Homer1b/c protein after CUMS+ SSDL ↑*Homer1* mRNA after CUMS ↓*Homer1* mRNA after CUMS+ FLU ↓*Homer1* mRNA after CUMS+ SDDH ↓*Homer1* mRNA after CUMS+ SDDL	Liu et al. [163]
**Nitric oxide synthases**						
**nNOS** **iNOS** **eNOS**	Wistar male rats	L-arginine 750 mg/kg (AR) (causes a depressive state) Fluoxetine 10 mg/kg i,p. (FLU) Milnacipran 30 mg/kg i,p. (MIL) Mirtazapine 10 mg/kg i,p. (MIR)	Cerebellum (CER) Frontal cortex (FC) Midbrain (MID) Olfactory bulb (OB) Pons Striatum Temporal cortex (TC) Thalamus (TH) Hippocampus (HP)	qPCR	↔N*os1* mRNA after AR ↔N*os1* mRNA after AR+ FLU, AR+MIL, AR+MIR—all brain regions ↑N*os2* mRNA after AR—CER, FC, MID, OB, pons, striatum, TC, TH ↔*Nos2* mRNA after AR—HP ↔N*os2* mRNA after AR+ FLU, AR+MIL, AR+MIR—all brain regions ↔N*os3* mRNA after AR ↑N*os3* mRNA after AR+ FLU—CER, Hp, midbrain, pons, striatum, thalamus, FC ↔N*os3* mRNA after AR+ FLU—OB, TC; after AR+ MIL and AR+ MIR—all brain regions	Yoshino et al. [178]
**nNOS**	Sprague Dawley male rats	CUS—28 days Memantine 10 mg/kg/day i,p. (MEM) 45 min before the application of the stressor, for 28 days	Hippocampus	Western blot	↑nNOS protein after CUS ↓nNOS protein after CUS + MEM	Mishra et al. [179]

***Abbreviations: CX717—****2,1,3-benzoxadiazol-5-yl(morpholin-4-yl)methanone; low-impact ampakine; **NSC23766**—(*N6*-[*2*-(*4*-Diethylamino-1-methyl-butylamino)-*6*-methyl-pyrimidin-4-yl]-2-methyl-quinoline-4,*6*-diamine), Rac1 inhibitor; **YY-21**—novel furostan skeleton secondary timosaponin obtained from timosaponin B-III via hydrolysis with hydrochloric acid; **qPCR**—quantitative polymerase chain reaction.*

## Data Availability

Not applicable.

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
