# Peer review of "Postsynaptic Proteins at Excitatory Synapses in the Brain—Relationship with Depressive Disorders"

_ijms, 2022, doi:10.3390/ijms231911423_

Round 1
Reviewer 1 Report
1. You must enhance your writing of the study's purpose to ensure that it is clear “Therefore, this review aimed to systematize the knowledge about changes in both the level of PSD proteins and post-translational modifications in patients with DDs and in animal models of depression and after drugs with antidepressant activity. Moreover, the expression of the genes encoding them was revised”.
2. The lack of references is clear, especially in the introduction part; you need to check each sentence and add its references; moreover, you must include newly published references in addition to the old ones to your review.
3. After mentioning the abbreviation for the first time, use it rather than the full term, such as “glutamate”.
4. In page 4,"'GluA2 is essential for correct AMPAR function because its deletion results in increased calcium permeability and intracellular blockade by polyamines"; Not just GluA2 deletion may result in a calcium-permeable receptor, but also the Q/R editing in GluA2 may also result in a calcium-permeable subunit.
5. In page 45, “In animals, induced stress caused a decrease in Homer-1a gene expression in the cortex [124] and proteins in the hippocampus, prefrontal cortex (in mice) [80] and increased in the prefrontal cortex (in rats) [129]”. You may improve this sentence.
6. These words must be edited throughout the manuscript: potenitiation, re-etrant, Immunochistochemistry, Morover, neutrophic, behawior, debressive, AMAR.
7. In page 45, “PSD proteins are also thought to be involved in the development of depressive-type disorders, including and synaptic plasticity”, this sentence is meaningless.
Author Response
We thank very much for reviewing our manuscript. We also thank you very much for any critical comments that will improve the quality of our manuscript. We have tried to apply to all of them. The article and all tables has also been seriously revised linguistically and grammatically. All corrections made are marked in yellow.
- You must enhance your writing of the study's purpose to ensure that it is clear “Therefore, this review aimed to systematize the knowledge about changes in both the level of PSD proteins and post-translational modifications in patients with DDs and in animal models of depression and after drugs with antidepressant activity. Moreover, the expression of the genes encoding them was revised”.
Response: As suggested by the Reviewer, we have verified and strengthened the description of the review’s purpose.
- The lack of references is clear, especially in the introduction part; you need to check each sentence and add its references; moreover, you must include newly published references in addition to the old ones to your review.
Response: As suggested by the reviewer, we have included additional references in the publication. A total of 49 (pos. 158-207 with References) new referees have been added.
- After mentioning the abbreviation for the first time, use it rather than the full term, such as “glutamate”.
Response: Of course we agree with the reviewer. Use of abbreviations should be consistent. The abbreviation should be entered in the text when the full name is first used. We complied with this remark. We tried to use the abbreviations consistently: Glu, AMPAR, NMDAR, CNS, PSD etc.
- In page 4,"'GluA2 is essential for correct AMPAR function because its deletion results in increased calcium permeability and intracellular blockade by polyamines"; Not just GluA2 deletion may result in a calcium-permeable receptor, but also the Q/R editing in GluA2 may also result in a calcium-permeable subunit.
Response: We have enriched this part of the text with the importance of the Q/R editing for the functioning of the GluA2 subunit and GluA2-containing AMPARs.
- In page 45, “In animals, induced stress caused a decrease in Homer-1a gene expression in the cortex [124] and proteins in the hippocampus, prefrontal cortex (in mice) [80] and increased in the prefrontal cortex (in rats) [129]”. You may improve this sentence.
Response: This sentence and many others in the text have been corrected to make them clearer and more understandable.
- These words must be edited throughout the manuscript: potenitiation, re-etrant, Immunochistochemistry, Morover, neutrophic, behawior, debressive, AMAR.
Response: We tried to identify and correct all linguistic mistakes in the text.
- In page 45, “PSD proteins are also thought to be involved in the development of depressive-type disorders, including and synaptic plasticity”, this sentence is meaningless.
Response: We agree with the Reviewer that this sentence was meaningless, hence we improved it.

Reviewer 2 Report
This is a review article which discussed the role of postsynaptic proteins in the excitatory synapse in detail and tried to organize the knowledge from published studies on humans and animals about changes that occur in the course of depression as well as after antidepressants.
Although the current knowledge of depression and molecular changes in response to this mental disorder is not necessarily successfully organized, the large amount of literature quoted in this review may be useful for future studies in this field.
This reviewer has no major objection against this manuscript but feels the language needed to improve somewhat.
Author Response
Reviewer 1
We thank very much for reviewing our manuscript. We also thank you very much for any critical comments that will improve the quality of our manuscript. We have tried to apply to all of them. The article and all tables has also been seriously revised linguistically and grammatically. All corrections made are marked in yellow.
- You must enhance your writing of the study's purpose to ensure that it is clear “Therefore, this review aimed to systematize the knowledge about changes in both the level of PSD proteins and post-translational modifications in patients with DDs and in animal models of depression and after drugs with antidepressant activity. Moreover, the expression of the genes encoding them was revised”.
Response: As suggested by the Reviewer, we have verified and strengthened the description of the review’s purpose.
- The lack of references is clear, especially in the introduction part; you need to check each sentence and add its references; moreover, you must include newly published references in addition to the old ones to your review.
Response: As suggested by the reviewer, we have included additional references in the publication. A total of 49 (pos. 158-207 with References) new referees have been added.
- After mentioning the abbreviation for the first time, use it rather than the full term, such as “glutamate”.
Response: Of course we agree with the reviewer. Use of abbreviations should be consistent. The abbreviation should be entered in the text when the full name is first used. We complied with this remark. We tried to use the abbreviations consistently: Glu, AMPAR, NMDAR, CNS, PSD etc.
- In page 4,"'GluA2 is essential for correct AMPAR function because its deletion results in increased calcium permeability and intracellular blockade by polyamines"; Not just GluA2 deletion may result in a calcium-permeable receptor, but also the Q/R editing in GluA2 may also result in a calcium-permeable subunit.
Response: We have enriched this part of the text with the importance of the Q/R editing
for the functioning of the GluA2 subunit and GluA2-containing AMPARs.
- In page 45, “In animals, induced stress caused a decrease in Homer-1a gene expression in the cortex [124] and proteins in the hippocampus, prefrontal cortex (in mice) [80] and increased in the prefrontal cortex (in rats) [129]”. You may improve this sentence.
Response: This sentence and many others in the text have been corrected to make them clearer and more understandable.
- These words must be edited throughout the manuscript: potenitiation, re-etrant, Immunochistochemistry, Morover, neutrophic, behawior, debressive, AMAR.
Response: We tried to identify and correct all linguistic mistakes in the text.
- In page 45, “PSD proteins are also thought to be involved in the development of depressive-type disorders, including and synaptic plasticity”, this sentence is meaningless.
Response: We agree with the Reviewer that this sentence was meaningless, hence we improved it.
Reviewer 2
We thank very much for reviewing our manuscript. We also thank you very much for any critical comments that will improve the quality of our manuscript. We have tried to apply to all of them. The article and all tables has also been seriously revised linguistically and grammatically. All corrections made are marked in yellow.
This reviewer has no major objection against this manuscript but feels the language needed to improve somewhat.
Response: As suggested by the Reviewer, the content of the review has been thoroughly verified and corrected in many places in terms of language and grammar.
